

# What are the challenges for modelling isoprene and monoterpene emission dynamics of subarctic plants?

Jing Tang[1,2], Guy Schurgers[2,3], Hanna Valolahti[1,2], Patrick Faubert[4], Päivi Tiiva[5], Anders Michelsen[1,2], Riikka Rinnan[1,2]

[1]Terrestrial Ecology Section, Department of Biology, University of Copenhagen, Copenhagen, Denmark

[2]Center for Permafrost, University of Copenhagen, Copenhagen, Denmark

[3]Department of Geosciences and Natural Resource Management, University of Copenhagen, Copenhagen, Denmark

[4]Chaire en éco-conseil, Département des sciences fondamentales, Université du Québec à Chicoutimi, 555, boulevard de l'Université, Chicoutimi, Québec, Canada, G7H 2B1

[5]Department of Environmental and Biological Sciences, University of Eastern Finland, PO Box 1627, 70211 Kuopio, Finland

*Correspondence to*: Jing Tang (Jing.Tang@bio.ku.dk)

**Abstract.** The Arctic is warming at twice the global average speed, and the warming-induced increases in biogenic volatile organic compounds (BVOC) emissions from arctic plants are expected to be drastic. The current global models' estimations of minimal BVOC emissions from the Arctic are based on very few observations and have been challenged by increasing field data. This study applied a dynamic ecosystem model, LPJ-GUESS, as a platform to investigate short-term and long-term BVOC emission responses to climate warming. Field observations in a subarctic heath tundra with long-term (13 years) warming treatments were extensively used for parameterizing and evaluating BVOC related processes. We proposed an adjusted temperature (T) response curve for arctic plants with much stronger T sensitivity than the commonly-used algorithms for large-scale modelling. The simulated emission responses to 2 °C warming between the adjusted and original T response curves in the model were evaluated against the observed warming responses (WR) at short-term scales. Moreover, the model's responses to higher levels' warming (4 °C and 8 °C) were also investigated as a sensitivity test. The model was able to reproduce vegetation $CO_2$ fluxes as well as day-to-day variability of isoprene and monoterpene emissions. The modelled BVOC WR, especially for isoprene, were better captured by using the adjusted T response curve, comparing with using the original one. A few days' underestimation of leaf T led to the underestimated emission rates as well as WR. During 1999-2012, the modelled annual mean isoprene and monoterpene emissions were 20 and 8 mg C m$^{-2}$ yr$^{-1}$, with an increase in emission by 55 % and 57 % for 2 °C summertime warming, respectively. Warming by 4 °C  and 8 °C further elevated isoprene emission for all years compared with 2 °C warming, but the impacts on monoterpene emissions levelled off because of a decreased coverage of monoterpene-emitters among the evergreen prostrate dwarf shrubs. The high WR captured by the adjusted T response curve highlight the strong T sensitivity of arctic plants. At short-term scale, the WR seem to be strongly





impacted by leaf T; while at long-term scale, the WR are a combined effect of plant functional type (PFT) dynamics as well as instantaneous BVOC responses to warming. The identified essential issues associated with estimating arctic BVOC emissions are: (1) leaf T estimation/extrapolation based on air T; (2) PFT parameterization accounting for BVOC emission features as well as PFT's responses to warming; and (3) representation of vegetation dynamics in the past and the future.

## 1 Introduction

Biogenic volatile organic compounds (BVOC) are reactive hydrocarbons mainly emitted by plants. Emissions of these secondary metabolites are involved in plants growth, plant defence against biotic and abiotic stresses, plant communication as well as reproduction (Laothawornkitkul et al., 2009; Peñuelas and Staudt, 2010; Possell and Loreto, 2013). BVOC synthesis is regulated by enzyme activity, and many compounds are emitted in a temperature (T)- and light density (Q)-dependent manner (Li and Sharkey, 2013). BVOC released into the atmosphere react with hydroxyl radicals (OH), which could reduce the atmosphere's oxidative capacity and therefore lengthen the lifetime of the greenhouse gas methane ($CH_4$) (Di Carlo et al., 2004; Peñuelas and Staudt, 2010). An increase in BVOC emission could also elevate the tropospheric ozone ($O_3$) concentration (Hauglustaine et al., 2005) as well as secondary organic aerosol (SOA) formation (Paasonen et al., 2013). Global estimates of non-methane BVOC emissions are in the range of 700-1000 Tg C $yr^{-1}$, of which isoprene and monoterpenes contribute most of the emissions (~70 % and 11 %, respectively (Sindelarova et al., 2014)). The modelled emission rates for isoprene are of similar magnitude as for $CH_4$ (Arneth et al., 2008). However, the current estimates of regional emission distributions are highly uncertain for both isoprene and monoterpenes for two reasons: 1) the current emission estimates are based on field studies mainly covering tropical, temperate and boreal ecosystems (Guenther et al., 2006), lacking observational data for the Subarctic and Arctic; 2) the uncertainties in driving variables (vegetation distribution, climate data, emission capacity) and in emission responses to these drivers (Guenther et al., 2006; Arneth et al., 2008). For instance, plants adapted to the cold environment of the Arctic appear to respond to warming differently than plants from low latitudes (Rinnan et al., 2014). Till now, the emissions from high latitudes (including the Arctic and the Subarctic) have been assumed to be minimal due to low foliar coverage, T and plant productivity (Guenther et al., 2006; Sindelarova et al., 2014). However, recent observations from the Arctic have indicated the need for revising the current assumption, as higher emissions than anticipated in large-scale models have been measured (Ekberg et al., 2009; Holst et al., 2010; Potosnak et al., 2013; Rinnan et al., 2014; Schollert et al., 2014). Furthermore, field experiments focusing on the effects of climate warming on BVOC emissions have found unexpectedly high responses of BVOC release to a few degrees of warming (Tiiva et al., 2008; Faubert et al., 2010; Valolahti et al., 2015; Lindwall et al., 2016a), which has underlined the potentially significant role of arctic BVOC emissions under changing climate. The Arctic is warming at approximately twice the global rate (IPCC, 2013) and the warming-induced drastic vegetation changes (AMAP, 2012) could impose substantial changes in BVOC emission.



Both isoprene and monoterpenes are produced through the 2-C-methyl-D-erythritol 4-phosphate/1-deoxy-D-xylulose-5-phosphate (MEP-DOXP) pathway and are reaction products of glyceraldehyde-3-phosphate (G3P) and pyruvate. G3P is produced along the chloroplastic Calvin Cycle and serves as the chief precursor. Mechanistic models have often linked the biosynthesis of isoprene and monoterpenes with photosynthesis processes (Niinemets et al., 1999; Martin et al., 2000;

Zimmer et al., 2003; Pacifico et al., 2011; Unger et al., 2013; Grote et al., 2014). In the short-term (hours-days), the responses to Q and T of isoprene and part of monoterpene productions are very similar to those of photosynthesis, but with a higher T optimum for BVOC production than photosynthesis (Guenther et al., 1995; Arneth et al., 2007). Furthermore, some monoterpenes can be emitted from storage pools in plant organs (e.g. glands or resin ducts, (Franceschi et al., 2005)). Along with the short-term responses, the long-term (days or longer) BVOC dynamics are affected by vegetation composition

changes (Faubert et al., 2011; Valolahti et al., 2015), seasonality (Staudt et al., 2000; Hakola et al., 2006), past weather conditions (Ekberg et al., 2009; Guenther et al., 2012) and vegetation growing conditions, e.g., soil water and nutrient availability (Possell and Loreto, 2013), atmospheric $CO_2$ (Wilkinson et al., 2009) and ozone levels (Loreto et al., 2004; Calfapietra et al., 2007). In comparison with empirical models (Guenther et al., 1995; Guenther et al., 2006; Guenther et al., 2012), process-based ecosystem models can generally capture more of these long-term effects and could thus be more useful

in terms of predicting long-term emission responses to environmental changes. Usually, estimates of BVOC responses to Q and T are based on the Guenther algorithm (referred here as G93, (Guenther et al., 1993)) and observed emission rates are often standardized to emission capacity at standard conditions (T of 30$^\circ$C and photosynthetically active radiation , PAR of 1000 μmol m$^{-2}$ s$^{-1}$) using the G93 algorithm to allow for comparison with other observations. However, Ekberg et al. (2009) found that the T response of the G93 algorithm is not sensitive enough to capture the observed high T responses of wet

tundra sedges, which was further supported by other studies in the high latitudes (Faubert et al., 2010; Holst et al., 2010). Furthermore, species-specific emission profiles (Rinnan et al., 2011; Rinnan et al., 2014; Schollert et al., 2015; Vedel-Petersen et al., 2015) have not yet been integrated into the modelling of (sub)arctic BVOC emissions (Arneth et al., 2011; Guenther et al., 2012; Sindelarova et al., 2014), which need to be included as a trait of plant functional types (PFTs), especially when targeting at studying drastic impacts of climate change on vegetation composition as well as BVOC

emissions in the Arctic. In addition, tundra plants with relatively dark surfaces and low growth form (commonly less than 5 cm tall) may experience much higher leaf T than the air T at 2 m height provided by weather stations (Körner, 2003; Scherrer and Körner, 2010; Lindwall et al., 2016a) which could lead to larger emissions than anticipated in current models.

The aim of this work was to integrate the observed emission features of arctic plants into a process-based ecosystem model in order to improve the current model estimations of arctic BVOC emissions, and to advance our understandings of the

emission dynamics for arctic ecosystems in a warming future. The process-based dynamic ecosystem model, LPJ-GUESS (Smith et al., 2001; Smith et al., 2014) was used as a platform to distill short-term and long-term responses of BVOC emissions to changes in climate for arctic plants. The model links isoprene and monoterpene productions with photosynthesis (Arneth et al., 2007; Schurgers et al., 2009). For the application to a subarctic heath tundra, the process parameterization utilized field observations of the long-term (13 years) warming treatment effects on vegetation composition





and BVOC emissions (Tiiva et al., 2008; Faubert et al., 2010; Valolahti et al., 2015). The specific objectives of this study were: (1) To capture the observed BVOC T sensitivity for a subarctic ecosystem; (2) To address short-term and long-term impacts of warming on ecosystem BVOC emissions; (3) To diagnose key model developments needed to better present BVOC dynamics for the arctic region.

## 2 Materials and methods

### 2.1 Study area and observational data

The data used in this modelling study were collected at a dwarf shrub/graminoids heath tundra site located in Abisko, northern Sweden (68°21'N, 18°49'E). The vegetation consists of a mixture of evergreen and deciduous dwarf shrubs, graminoid and forbs. A long-term field experiment was established at this site in 1999 to investigate the effects of climate warming and increasing litter fall, resulting from the expanding tundra vegetation on the functioning of the ecosystem. The experiment included control (C), warming (W), litter addition (L) and a combined warming and litter addition (WL) treatments (Rinnan et al., 2008). In the current study, we focused only on the observations from the C and W treatments. Each treatment, covering an area of $120 \times 120$ cm, was replicated in six blocks. The W treatments used open-top chambers (OTCs), which passively increased air T by around 2 °C, and also caused a 10 % reduction in PAR (Valolahti et al., 2015).

During the years 2006, 2007, 2010 and 2012, BVOC emission rates were measured for all plots by sampling air from transparent polycarbonate chambers into adsorbent cartridges using a push-pull enclosure technique and analysis by gas chromatography-mass spectrometry (GC-MS). The enclosure covered a $20 \times 20$ cm area in each plot. The isoprene emission datasets for 2006-2007 can be found in Tiiva et al. (2008) and for 2012 in Valolahti et al. (2015). For the year 2010, isoprene emission rates were not analyzed due to technical problems (Valolahti et al., 2015). The monoterpene emissions datasets have been published by Faubert et al. (2010) for 2006-2007 and by Valolahti et al. (2015) for 2010 and 2012. Notably, BVOC in this study only refers to isoprene and monoterpenes. Closed chamber-based $CO_2$ fluxes were measured in the same area during the same years (Tiiva et al., 2008; Valolahti et al., 2015). Species composition and coverage in the plots in the same years were estimated by point intercept-based method, in which a hit is recorded each time a plant species is touched by a pin lowered through 100 holes covering the plot area of $20 \times 20$ cm (Tiiva et al., 2008; Valolahti et al., 2015). Species composition was measured in June for 2006, 2010 and 2012, and in June, July and August for the year 2007. Furthermore, air T and PAR inside the chamber were also monitored during the BVOC sampling time.

### 2.2 LPJ-GUESS

#### 2.2.1 LPJ-GUESS general framework

LPJ-GUESS is a climate-driven dynamic ecosystem model with process-based representations of plant establishment, mortality, disturbance and growth as well as soil biogeochemical processes (Smith et al., 2001; Sitch et al., 2003).



Vegetation in the model is defined and grouped by plant functional types (PFTs), which are based on plant phenological and physiognomic features, combined with bioclimatic limits (Sitch et al., 2003; Wolf et al., 2008). The model has been widely and successfully applied for simulating vegetation and soil carbon fluxes as well as vegetation dynamics at different spatial scales (Wolf et al., 2008; Hickler et al., 2012; Smith et al., 2014; Tang et al., 2015). In LPJ-GUESS, a generalized Farquhar photosynthesis model (Farquhar et al., 1980) for large-scale modelling is used to simulate carbon assimilation (Collatz et al., 1991; Haxeltine and Prentice, 1996b; Haxeltine and Prentice, 1996a) and the generalized model is built on the assumption of optimal nitrogen (N) allocation in the vegetation canopy (Sitch et al., 2003). Daily net photosynthesis is estimated using a standard nonrectangular hyperbola formulation, which gives a gradual transition between the PAR-limited ($J_E$) and the Rubisco-limited ($J_C$) rates of assimilation (Haxeltine and Prentice, 1996b). For $C_3$ plants, $J_E$ is a function of the canopy absorbed PAR, the intrinsic quantum efficiency for $CO_2$ uptake ($\alpha_{c3}$), the $CO_2$ compensation point ($\Gamma^*$) and the internal partial pressure of $CO_2$ ($p_i$) (Collatz et al., 1991; Haxeltine and Prentice, 1996b). $J_C$ is related to the maximum catalytic capacity of Rubisco per unit leaf area ($Vm$), $\Gamma^*$, $p_i$ and the Michaelis-Menten constant for $CO_2$ ($K_C$) and $O_2$ ($K_O$).

### 2.2.2 BVOC modelling

In LPJ-GUESS, isoprene (Arneth et al., 2007) and monoterpene (Schurgers et al., 2009) emissions are simulated as a function of the photosynthetic electron flux used for photosynthesis. The productions ($I$) of isoprene and monoterpenes are computed as

$$I = \alpha J \varepsilon \text{, where } \alpha = \frac{C_i - \Gamma^*}{6 \times (4.67 C_i + 9.33 \Gamma^*)} \tag{1}$$

where $J$ is the rate of photosynthetic electron transport, $\alpha$ converts from photon fluxes into terpenoid units and $C_i$ is the leaf-internal $CO_2$ concentration without water stress. The synthesis of both compounds is linked to the photosynthetic electron transport (Niinemets et al., 1999; Niinemets et al., 2002). A fraction ($\varepsilon$) of the electron transport contributing to terpenoid productions (Eq. 2) is determined from a plant-specific fraction under standard conditions ($\varepsilon_S$, usually at a T of 30 °C and a PAR of 1000 µmol m$^{-2}$ s$^{-1}$) which is adjusted for leaf T, leaf phenology ($\sigma$), and atmospheric $CO_2$ concentration:

$$\varepsilon = f(T) f(\sigma) f(CO_2) \varepsilon_S \tag{2}$$

The standard fraction $\varepsilon_S$ is computed from the often reported standard emission rate (emission capacity) together with the simultaneously estimated photosynthetic electron flux under these standard conditions in the model. The T response corrects for the T optimum from terpenoid synthesis, which is higher than that for photosynthesis:

$$f(T) = e^{\alpha_\tau (T - Ts)} \tag{3}$$

The parameter $\alpha_\tau$ represents the T sensitivity and $T_S$ is often 30 °C. However, for the study in the Subarctic, the used reference $T_S$ as standard conditions and the T responses were adjusted based on the observation data and will be discussed below. The seasonality function, $f(\sigma)$, only applies to isoprene production and is based on the work by Arneth et al. (2007)




which was later modified by Schurgers et al. (2011). The atmospheric $CO_2$ concentration enhances terpenoid synthesis when the concentration is lower than ambient, and vice versa, which are represented by the function $f(CO_2)$.

For monoterpenes, a storage pool ($m$) is assigned to represent the specific (long-term) storage of monoterpenes within a leaf (Schurgers et al., 2009). The storage pool is only implemented for coniferous and herbaceous PFTs (see Table S1). The emission of monoterpenes from the storage is a function of T and $m$ with an average residence time ($\tau$). $\tau_s$ is the residence time at the standard T of 30 °C (adjusted to 20 °C in this study, consistent with the modification on the T responses of terpenoid synthesis). The residence time $\tau$ is adjusted based on the standard condition $\tau_s$ for T responses with a $Q_{10}$-relationship.

$$M_{emis} = m/\tau$$

$$\tau = \frac{\tau_s}{Q_{10}^{(T-T_s)/10}}$$

(4)

In LPJ-GUESS, the BVOC response to light resides in the photosynthesis processes (light-dependence of $J$ in Eq. 1). Additionally, considering the high sensitivity of BVOC to leaf T, Schurgers et al. (2011) further extended the computation of leaf T from air T to model BVOC emission. The calculation of leaf T in the model was based on solving leaf energy balance, where the incoming shortwave and longwave radiation is balanced by the outgoing longwave radiation and sensible heat fluxes as well as latent heat loss. For this study, the representation of incoming longwave radiation was modified by specifically considering the cloud emission of longwave radiation relative to clear-sky condition (Sedlar and Hock, 2009).

### 2.3 Simulation setup

#### 2.3.1 Input data

The daily climate data of air T, air T range and precipitation for the period 1984-2012 (Callaghan et al., 2013; Tang et al., 2014) were provided by the Abisko scientific research station (Abisko Naturvetenskapliga Station, ANS). Four gaps in daily radiation data from ANS (during the periods of 1984/01/01-1984-06-30, 2006/06/09-2006/06/16, 2007/02/13-2007/02/15, 2011/07/23-2011/08/17) were filled by the Princeton reanalysis dataset (Sheffield et al., 2006) for the grid cell nearest Abisko. The annual $CO_2$ concentrations for the whole study period (1984-2012) were obtained from McGuire et al. (2001) and TRENDS (http://cdiac.esd.ornl.gov/trends/co2/contents.htm).

#### 2.3.2 Plant functional types

The dominant plant species from the observations (Valolahti et al., 2015) were divided into 7 PFTs (Table 1). The PFT parameters (see Table S1) were mainly derived from previous studies for the arctic region using LPJ-GUESS (Wolf et al., 2008; Miller and Smith, 2012; Tang et al., 2015), but they also extend the arctic PFT lists to consider BVOC emission characteristics. The low summergreen shrubs (LSS) were divided into a *Salix*-type (SLSS; high isoprene emitter) and a non-*Salix*-type (NSLSS; e.g., *Betula nana*-dominance, predominantly monoterpenes rather than isoprene emitters) (Schollert et





al., 2014; Vedel-Petersen et al., 2015). Furthermore, due to the abundance of prostrate dwarf shrubs (PDS) in the study area, distinguishing PDS (canopy height lower than 20 cm) from low shrubs (canopy height lower than 50 cm) was implemented through adjusting parameters controlling vegetation height. The PDS-type was further divided into two PFTs with evergreen and deciduous phenology. Moss, widely appeared in the study area, was not distinguished from forbs and lichens, due to

limited data for parameterizing moss physiognomic features and their preferable growing conditions. In LPJ-GUESS, the canopy of each tree is divided into thin layers (original value is 1.0 m in a forest canopy) in order to integrate PAR received by each tree. The depth of this layer was reduced to 10 cm in this study to better capture the vertical profile of low and prostrate shrubs. In addition, the original specific leaf area (SLA, $m^2$ kg $C^{-1}$) values in LPJ-GUESS were estimated based on a fixed dependency on leaf longevity (Reich et al., 1997). In our study, a fixed SLA was assigned to each PFT (Oberbauer

and Oechel, 1989) to improve the simulated leaf area index (LAI) for arctic plants. Emission capacities for the PFTs were determined from available leaf-level measurement data from the Subarctic and Arctic. The details about the data sources for parameterizing emission capacity at 30 °C ($I_{S30}$) and 20 °C ($I_{S20}$) can be found in Table S2 and the averaged emission capacities (among all literature data in Table S2) for each PFT as well as the representative plant species can be found in Table 1. The emission rates from the literature are generally provided as standardized emission capacities at 30 °C using G93

algorithm and these values were further rescaled to 20 °C using the adjusted T response curve (Fig. 1).

### 2.3.3 Model calibration and validation

The modelled $CO_2$ fluxes, LAI as well as the BVOC T response were firstly calibrated before evaluating the modelled daily BVOC emission rates. Two (from the year 2006 and 2007) out of four years' closed chamber measured net ecosystem production (NEP), ecosystem respiration (ER) and the estimated gross primary production (GPP) as well as point intercept-

based species composition were used for calibrating and the rest two years' data (from the year 2010 and 2012) were used for validating the simulated carbon cycle processes. Previous studies focusing on light responses of NEP for arctic plants (Shaver et al., 2013; Mbufong et al., 2014) have reported relatively low quantum efficiencies ($\alpha_{c3}$) caused by overall low sun angle conditions and low leaf area. A thorough sensitivity study of parameters used in LPJ-GUESS (Pappas et al., 2013) has found that $\alpha_{c3}$ is the most influential parameter in terms of the simulated vegetation carbon fluxes. Also, a pre-validation of

the modelled $CO_2$ fluxes with the observations in this study using the default $\alpha_{c3}$ value (0.08) has found a large overestimation of both GPP and ER (not shown). Therefore, a sampling of $\alpha_{c3}$ (using the range of 0.02 to 0.125 μmol $CO_2$ μmol photons$^{-1}$, proposed by Pappas et al. (2013)) was conducted to find the best value to depict the observed GPP, ER and LAI of the year 2006 and 2007 for the subarctic ecosystem (Fig. S1). The evaluation of the calibrated $\alpha_{c3}$ on the simulated $CO_2$ fluxes and vegetation composition was conducted using the observed $CO_2$ fluxes and the point intercept-based plant

coverage data from 2010 and 2012, respectively.

The daytime air T in the study area is often below 20 °C (Ekberg et al., 2009), and standardization of terpenoid emissions to 20 °C, instead of 30 °C, has been suggested for modelling in boreal and arctic ecosystems (Holst et al., 2011, Ekberg et al., 2009) due to plants' adaptation to low T environment. In the model, the photosynthetic electron fluxes under standardized



conditions are simulated in order to convert the input emission capacity to the standard fraction ($\varepsilon_S$, see Eq. 2). The choice of the standardized T will influence the estimated fraction of electron fluxes for BVOC synthesis. In this study, a data fitting to the suggested standard T of 20 °C was conducted using the observed ecosystem-level isoprene emission rates in July together with measurement chamber air T from the control plots. The observations were mostly conducted during daytime with

relatively high PAR values, and therefore the response of the emission rates to light was not specifically considered in the current data fitting. Potential feedbacks from the variations in the atmospheric $CO_2$ concentration were ignored for the three years with isoprene sampling. The data collected from different blocks were separated for the curve fitting and the parameters controlling T response ($\alpha_\tau$ in Eq. 3) were determined (Fig. 1). An adjusted $\alpha_\tau$ value of 0.23 was chosen after fitting all the data from July over three years' measurements. Apart from the low $R^2$ value for block 1, the data were well

captured by the exponential shape ($R^2 > 0.8$) of the T response curve. The calibrated T responses were used for standardizing leaf-level emission rates (see $I_{S20}$, Table 1) as well as estimating emission rates in the model.

After calibrating the modelled carbon cycle processes as well as the $\alpha_\tau$ value for BVOC T response, the vegetation composition was evaluated using simulated LAI against the point intercepted-based species composition. The species were grouped into the corresponding PFTs for comparison and the point intercept-based hits within the same PFT group were

summed. The summed hits were divided with 100 pin hits to compare with the modelled LAI. After evaluating the modelled $CO_2$ fluxes and LAI, the modelled isoprene and monoterpene emission rates were compared with the observations. The simulated emissions, represented as daily average values ($\mu g\ C\ m^{-2}\ h^{-1}$, daily emission rates divided by day length) may not allow accurate comparisons with the actual emission rates which were typically conducted in the middle of the day (between 10 am – 2 pm). Therefore, the emission rates at noon were estimated by an additional computation of the emission, applying

the leaf T computed from the daily maximum air T for photosynthesis and BVOC emissions. Due to lacking of data about the daily maximum PAR value, the daily average PAR was used for estimating the emission rates at noon. We expect the emission rates at noon to be more comparable with the observations than the daily mean emissions, considering diurnal dynamics of emissions.

### 2.3.4 Warming experiment

To simulate the observed warming responses from the OTCs, warming of 2 °C was imposed in the model for the growing season (the period with OTC warming) (Tiiva et al., 2008; Valolahti et al., 2015). The modelled warming responses (WR, emission differences between C and W treatments) using the original T response ($\alpha_\tau = 0.1$, $T_S = 30$ °C, Eq. 3) and the adjusted T response ($\alpha_\tau = 0.23$, $T_S = 20$ °C, Eq. 3 and Fig. 1) were compared with the observed WR. Furthermore, apart from the settings with 2 °C warming, additional simulations with a warming of 4 °C and 8 °C, reflecting the range of climatic

projections in this region (IPCC, 2013), were also conducted to test the anticipated ecosystem-scale responses of different levels of warming.



## 3 Results

### 3.1 Modelled CO$_2$ fluxes and vegetation composition

The parameter $\alpha_{c3}$, determining the efficiency in converting solar radiation to carbon, showed strong impacts on the modelled ecosystem CO$_2$ fluxes and LAI for the growing seasons 2006 and 2007 (Fig. S1). For CO$_2$ fluxes, the lowest root mean
square error (RMSE) values occurred at 0.035 μmol CO$_2$ μmol photons$^{-1}$ for GPP and ER, while the lowest RMSE value for LAI was 0.051 μmol CO$_2$ μmol photons$^{-1}$. A value of 0.04, consistent with the study by Shaver et al. (2013) was selected for $\alpha_{c3}$ to limit the RMSE values of both modelled CO$_2$ fluxes and LAI. Using this value for $\alpha_{c3}$ calibrated with observation data from 2006 and 2007, the model captured the observed day-to-day variations as well as magnitudes of the chamber-based GPP, ER and NEP from 2010 and 2012, though with a slight overestimation of CO$_2$ fluxes (particularly for the early growing
seasons, Fig. 2), and an underestimation of LAI (Fig. 3). For the year 2012, the model showed large overestimations of the observed GPP and ER for the limited number of measurements in this growing season.

The details about the representative species for each PFT can be found in Table 1. For the 5 groups of PFTs, the modelled growing season LAI values for the year 2010 and 2012 were generally lower than the point intercept-based coverage estimations from the field observations (note different left and right axes scales in Fig. 3), except for the *Salix*-type
summergreen shrubs and deciduous prostrate dwarf shrubs (SLSS+SPDS). The two most dominant vegetation groups in the C plots, forbs/lichens and evergreen shrubs, were captured by the model. However, the large coverage of graminoids (GRT) and non-*Salix*-type deciduous shrubs (NSLSS) was largely underestimated by our model.

In response to 2 °C warming, the modelled LAI for the shrub PFTs (SLSS+SPDS, NSLSS, LSE+EPDS) showed an increase, while the modelled LAI for graminoids and forbs/lichens largely decreased. Based on the observations, vegetation changes
have been to the same directions for two groups of shrubs (NSLSS and LSE+EPDS). However, an observed large increase of the coverage of forbs/lichens as well as a decreased coverage of graminoids in the W treatments for the year 2010 and 2012 were not captured by the model.

### 3.2 Modelled BVOC

#### 3.2.1 Seasonal variations

BVOC emissions are closely linked to leaf as well as ecosystem developments and simulating dynamic vegetation enables us to assess the model's performance in representing short-term emission changes in response to T and PAR, as well as long-term changes in vegetation development and distribution. The span of the BVOC measurements/samplings covered the main growing seasons over four years. The modelled emission rates in the C plots showed pronounced day-to-day as well as seasonal variations (Fig. 4). The modelled emissions of isoprene and monoterpenes were low in the spring and autumn, and
peaked on warm days during the summer. The day-to-day variations in the emissions agreed well with the variations of T and PAR. When both T and PAR were high through the growing season, the peaks of both isoprene and monoterpene



emissions occurred and the deviations between the modelled daily average and noon emissions became larger. The observed magnitude of isoprene emissions during daytime showed large spatial variations between the blocks for the days with the observed high average emission rates (blue error bars in Fig. 4) and the observed average rates (blue squares) were well captured by the modelled noon emissions. For monoterpene emissions, the modelled daily average was closer to the observation, especially for the years with generally low emissions (2006, 2010 and 2012). The observed high monoterpene emissions for a few days were relatively better captured by the modelled noon emission. The emissions of monoterpenes remained more constant towards the end of the growing season (not fully represented here).

### 3.2.2 Daily emissions

- Emission rates at the control conditions

The observed daily variations in isoprene emissions were mostly captured by the model (Fig. 4, Fig. 5). The observed isoprene emission rates (Fig. 5a) lay between the modelled daily average and daily noon emission rates, with the exception of a few days with much higher emission rates from the observations than the modelled noon emissions (22 August 2006, 10 July 2007 and 05 August 2007). For these dates, the observed chamber air T were higher than the modelled daily noon leaf T (squares in Fig. 5c) and were also higher than or close to 20 °C. For 06 July 2007, when the simulated noon leaf T was higher than 20 °C, the model captured the observed high emissions well. Notably, the model used air T at 2 m height from the ANS station to extrapolate the leaf T, while the measured T is the air T inside the measurement chamber. Over three growing seasons, the observed air T inside the chambers was on average 7.2 °C warmer than the modelled daily average and 3.4 °C warmer than the modelled daily noon leaf T. The modelled daily average, daily noon and the observed daytime emission rates were 9.1, 25.8 and 25.5 µg C m$^{-2}$ h$^{-1}$, respectively (all numbers averaged for the days on which measurements were made) and the modelled daily noon isoprene emission rates demonstrated better representation of the observed daytime emission rates.

For monoterpenes, the modelled daily average emission rates in the C plots (light grey bars in Fig. 6a) showed relatively closer values to the observations, compared to the modelled noon emission rates (dark grey bars in Fig. 6a). Over four sampling growing seasons , the modelled daily mean, daily noon and observed daytime emission rates were 2.4, 7.9 and 2.5 µg C m$^{-2}$ h$^{-1}$, respectively (all numbers averaged for the days on which measurements were made). The modelled daily mean showed better agreements with the observed low monoterpene emissions for 2006, 2010 and 2012, but underestimated the observed high emission rates for the year 2007. In 2007, the highest emission rates observed on 06 July were not captured by the modelled daily average, but were of similar magnitude as the modelled daily noon emission rates (with certain overestimations). Whereas the observed emissions showed great variations between years (1.3, 8.1, 0.3, 0.5 µg C m$^{-2}$ h$^{-1}$, for the four years measured, respectively), the simulated daily noon emissions were more similar between years (2.2, 3.0, 2.1 and 2.4 µg C m$^{-2}$ h$^{-1}$, respectively).

- Emission responses to 2 °C warming



In response to 2 °C warming, the modelled leaf T increased on average by 2 °C, while the observed chamber air T in the W plots increased 1.8 °C relative to the C plots on an average of four growing seasons with observations. For isoprene, the modelled WR (Fig. 5b) were relatively lower than the observed WR, especially for a few days with strong observed WR. Averaging over three years, the modelled daily average, noon and the observed WR were 5.7, 15.2 and 29.3 µg C m$^{-2}$ h$^{-1}$, respectively and warming increased the modelled daily average isoprene emission rates by 63 %, the daily noon by 59 % and the observed emissions by 115 % (all numbers averaged for the days on which measurements were made). Over three years, the observed strong WR for a few dates (e.g., 22 August 2006, 10 July 2007, 05 August 2007 and 14 June 2012) were underestimated by the modelled noon WR when the observed chamber air T in the C was close to or higher than 20 °C, but the modelled leaf noon T was below this level. However, for the day when both daily leaf T and chamber air T were over 20 °C (e.g., 13 June 2006, 06 July 2007), the observed WR were relatively higher than the modelled daily average, but lower than the modelled daily noon WR.

For monoterpenes, the modelled daily average, noon and the observed WR were 2.0, 6.0 and 2.5 µg C m$^{-2}$ h$^{-1}$, respectively. The averaged WR from the model noon emissions were much higher than the observations. The modelled daily average WR showed better agreement with the observations. For one day with extremely high WR (06 July 2007), the modelled noon WR better captured the strong responses. Averaging over four growing seasons, warming increased the modelled daily average monoterpene emissions by 81 %, the daily noon emission by 76 % and the observed emissions by 98 %.

The modelled daily noon WR using the adjusted BVOC T response ($\alpha_\tau = 0.23$, $T_S = 20$ °C, Eq. 3) were further compared with the simulation using the original T response ($\alpha_\tau = 0.1$, $T_S = 30$ °C, Eq. 3). For isoprene (Fig. 7a), the simulation using the adjusted T response showed a substantial increase of the modelled WR as well as a better agreement to the observations than the simulation using the original T response. The modelled WR using the original T response showed limitation in capturing the observed high WR. Averaging through three years, the modelled isoprene WR using the original T response only represented 11 % of the observed WR, while the modelled WR using the new T response captured 52 % of the observed WR. For monoterpenes, the modelled WR using the original T response showed relatively closer values to the observations for the years with the observed low WR (2006, 2010 and 2012). For the year 2007, the observed high monoterpene WR were better captured by the simulated WR with the new T response.

### 3.2.3 Annual emissions

A comparison of the simulated annual BVOC emissions from the C and W treatments demonstrated that the 2 °C warming during the growing seasons increased both isoprene and monoterpene annual emissions. Averaging over 13 years, the warming by 2 °C during the growing seasons increased annual isoprene and monoterpene emissions by 55 % and 57 %, respectively ($p < 0.01$, Mann-Whitney test). The modelled emissions showed strong inter-annual variations in response to warming (Fig. 8). For the warmest year (2011), the W treatment increased annual isoprene and monoterpene emissions by 99 % and 94 %, respectively. The mean annual isoprene and monoterpene emissions in the C for 1999-2012 were 20 and 8 mg C m$^{-2}$ yr$^{-1}$, respectively. For the four years with BVOC sampling, the modelled average WR were 58 % and 70 % for



annual isoprene and monoterpene emissions, respectively. The modelled WR at the annual scale were at the similar magnitudes as the modelled daily average WR for the days with BVOC samplings (63 % for isoprene and 81 % for monoterpenes).

The simulations imposing the warming by 4 °C or 8 °C during the same period as the 2 °C warming increased annual isoprene emissions by 120 % and 247 %, respectively ($p < 0.01$, Mann-Whitney test) and annual monoterpene emissions by 87 % and 167 %, respectively ($p < 0.01$, Mann-Whitney test). For isoprene, the strongest WR of all levels of warming appeared in 2011. Higher levels of warming further elevated isoprene emissions for all years, but the impact on monoterpene emissions levelled off for a decreased coverage of evergreen prostrate dwarf shrubs (EPDS) with 8 °C warming. The decrease in coverage of EPDS only occurred for the last few years with 4 °C warming. The different levels of warming generally increased shrub growths, but largely decreased the coverage of forbs/lichens and graminoids (CLM and GRT) (data not presented).

## 4 Discussion

### 4.1 Emission rates

The model was able to reproduce vegetation $CO_2$ fluxes (Fig. 2) as well as the main dynamics of isoprene and monoterpene emissions (Fig. 4), in spite of deficiencies in the representation of the observed vegetation composition (Fig. 3). The mismatch between the modelled LAI and the point intercepted-based vegetation coverage may be caused by an underestimation of the allocation of assimilated carbon to foliage in LPJ-GUESS and/or too low SLA values (Table S1). In LPJ-GUESS, the carbon allocation among different living tissues follows four allometric equations (see Eqs. 1-4 in Sitch et al. (2003)) to control the structural development of each modelled plant individual. The allometric parameters for some of the arctic PFTs used in this study were validated by Wolf et al. (2008) derived for a model applying a quantum efficiency $\alpha_{c3}$ of 0.08 at the regional scale, which may require further justification after the reduction in $\alpha_{c3}$ that was applied here to match the observed daily $CO_2$ fluxes. The reduced quantum efficiencies reflect the growth environment with low T as well as low sun angle in high latitudes (Shaver et al., 2013), but more observations are still needed to better quantify light use efficiency of arctic plants (Dietze et al., 2014). Furthermore, Van Wijk et al. (2005) found a close linkage between total foliar nitrogen (N) content and LAI for arctic plants, which was further supported by Campioli et al. (2009) for an arctic ecosystem dominated by *Cassiope tetragona*. However, the current simulations neither include C-N interactions nor consider potential impacts of N limitations on plants developments (Smith et al., 2014), which need to be improved in future model simulations in this region (Michelsen et al., 2012). The subdivision of arctic PFTs into smaller groups to specifically consider isoprene and monoterpene emission features has shown the importance for capturing the emission dynamics in this heath tundra ecosystem, but the development of arctic PFTs also poses challenges to consider the phenological and physiognomic features of mosses (currently aggregated in the CLM-type PFT, Table S1), which may bring additional uncertainties to the modelled LAI. At the same time, the current evaluation of the modelled LAI with the point intercept-based measurements of plant





coverage cannot disregard uncertainties from the field method itself, such as subjective judgement of species from each hit, and the potential influence from sampling inclining angles (Wilson, 2011). Also, the seasonal variation in leaf development as well as the randomly selected blocks from the heterogeneous landscape may further complicate the comparison of the simulated LAI with the local observations. Capturing the start of the growing season in the model is also crucial for

depicting the dynamics of seasonal $CO_2$ fluxes (Tang et al., 2015). The overestimated GPP in the beginning of growing seasons (Fig. 2a) suggests uncertainties in modelling the time of this start. The current algorithm for detecting start of growing season in large scale applications (Sykes et al., 1996) may not be sensitive enough for prediction of budburst of arctic plants (Pop et al., 2000).

The modelled annual isoprene and monoterpene emissions, 20 and 8 mg C m$^{-2}$ yr$^{-1}$ for 1999-2012, correspond to less than

0.1 % of the modelled GPP. The modelled emission rates are not only linked to the modelled photosynthesis fluxes, but also to the emission capacity assigned for each PFT (see Table S2). For some PFTs (e.g., the *Salix*-type and prostrate summergreen shrubs, SLSS and SPDS), the emission capacities in Table 1 are of similar magnitude as observed values that are applied in large-scale models for boreal forests (see Table 2 in (Rinne et al., 2009)). The relatively low observed emissions in comparison with lower latitudes are mainly caused by low T and plant biomass, and not by low emission

capacities (Holst et al., 2010).

Relative to isoprene emission, the magnitude of monoterpene emissions was much lower since the species in the study area were mostly considered to be isoprene emitters (Tiiva et al., 2008; Faubert et al., 2010). The model showed certain limitations in representing the observed low monoterpene emission rates (mainly for the year 2010 and 2012), which could be attributed to the prescribed value for splitting the produced monoterpenes into direct emissions (50 %) and emissions

from storage pools (50 %) (Schurgers et al., 2009). This split determines the distribution of monoterpene emissions over the year, and an allocation of the monoterpenes into storage pools results in a more gradual distribution of emissions. At the same time, the overestimated monoterpene emissions during the evaluating periods (mainly growing seasons) may also indicate the implemented storage residence time is too short and/or the temperature dependence of monoterpene emissions from the storage pool (Eq. 4) is too strong for arctic plants. The current observations of BVOC emissions only covered the

main growing season. Sampling over a longer season (Holst et al., 2010) could help to improve the parameterizing of this portioning, as well as the T response of emission rates from storage pools. Furthermore, ongoing $^{13}$C labeling experiment focusing on arctic mesocosm (Lindwall et al., unpublished data) could also help to identify the fractions of monoterpene emissions from production or storage. Additionally, evaluating the modelled daily emission rates with the field observed rates at certain period of a day cannot avoid potential impacts from the diurnal dynamics of BVOC emissions.

**4.2 Responses to warming**

The modelled increases of shrub coverage in response to the W treatment mostly followed the field observations (Valolahti et al., 2015) and are consistent with the general trend from other arctic studies (Wahren et al., 2005; Elmendorf et al., 2012). However, the observed increase of bryophytes is rather site-specific, which was not captured by the model. In contrast, the



modelled W-induced decreased coverage of graminoids and forbs/lichens agrees well with the large-scale trend identified by Elmendorf et al. (2012) who conducted a global synthesis of 61 tundra warming experiments. The decreasing soil moisture in W treatments (excluding wet ecosystems) is one of the main constraints on bryophyte coverage (Lang et al., 2012). Along with vegetation community responses, the short-term T responses of the vegetation are central for accurately depicting daily

BVOC responses to the warming treatment. Through enhancing the BVOC T sensitivity (from $\alpha_\tau = 0.1$, $T_S = 30\,°C$ to $\alpha_\tau = 0.23$, $T_S = 20\,°C$ in Fig. 1), the simulated BVOC WR (63 % for isoprene and 81 % for monoterpenes) became comparable to the observed responses (115 % for isoprene and 98 % for monoterpenes). The underestimation of a few days' strong isoprene WR could be partly attributed to the lower leaf T estimations derived from 2 m air T measured at the ANS station, than the observed daytime chamber air T in the low canopy (Fig. 5c). The low-statured plants in dry to mesic tundra ecosystems

(Schollert et al., 2014; Lindwall et al., 2016b) are efficient in absorbing heat and thus prone to have a high canopy T on a sunny day. This can directly elevate BVOC emissions as well as WR (Lindwall et al., 2016a). The observed strong WR can also be due to the potential side effects of the OTCs in the W treatment, e.g., reduced wind speed (De Boeck et al., 2012) and increased frequency of high-temperature events (Bokhorst et al., 2013). At annual to decadal timescales, the warming in the experimental plots caused changes in total plant biomass and species coverage which were found to contribute to the

increase in BVOC emissions after 13 years of treatments (Valolahti et al., 2015). These indirect effects on BVOC emissions were not yet identified for the years 2006 and 2007 (after 7-8 years of treatments) (Tiiva et al., 2008; Faubert et al., 2010), which points out the importance of accurately representing the temporal dynamics of vegetation as a driver of BVOC emissions. The modelled annual emissions in response to different degrees of warming (Fig. 8) have clearly elucidated the combined effects from the direct responses to summer warming as well as the indirect responses from vegetation changes.

The adjusted T response curve better represents subarctic plants' responses to warming than the original curve which has been parameterized for global simulations (Fig. 7). It further supports the earlier suggested stronger T sensitivity of BVOC emissions from arctic plants compared to plants from other regions (Ekberg et al., 2009; Holst et al., 2010; Rinnan et al., 2014). The commonly-used T response in Guenther's algorithm (Guenther et al., 1993) is based on the Arrhenius–type dependence of enzyme activities with an optimum T around 40 °C and the shape of the Guenther's response is very close to

the exponential curve with $\alpha_\tau$ value of 0.13 (using standard T of 30 °C) when leaf T is lower than 30 degrees. The high $\alpha_\tau$ value found in this study indicates that a slight T increase during summertime could cause a large increase of isoprene and monoterpene emissions from the studied cold subarctic ecosystem (Faubert et al., 2010; Holst et al., 2010). The pronounced high T responses in this tundra ecosystem further indicates that T responses of BVOC emissions could vary spatially (Niinemets et al., 2010) and points out the importance of applying locally optimized parameters for global estimates (Ekberg

et al., 2009).

### 4.3 Suggestions for other models and potential measurements

For extrapolating the current model developments to large-scale (regional) applications, we suggest to address the following issues: 1) The emission responses to T of arctic plants could be further tested based on laboratory experiments in controlled



conditions; 2) The strong decoupling of leaf T from air T and strong dependence of BVOC emissions on leaf T (Lindwall et al., 2016a) point to a need to capture leaf T accurately in models. Long-term parallel observations of both leaf and air T will be useful for the algorithm developments focusing on arctic vegetation (Rinnan et al., 2014); 3) The subdivision of the existing PFTs into groups featuring isoprene and monoterpene emissions are encouraged for other relevant modelling studies

(Grote et al., 2014), though additional data may be required for characterizing the new subgroups, such as bioclimatic limitations; 4) The potential impacts of seasonal dynamics of vegetation as well as phenology on emission capacities should be further identified with whole-season BVOC sampling (Staudt et al., 2000); 5) Arctic PFT's responses and/or acclimation to warmer climate should be better parametrized in the model to better represent long-term vegetation effects on BVOC emissions.

**5 Conclusions**

This study has demonstrated the model's ability to depict the observed isoprene and monoterpene emission rates as well as daily variations in the emission of a subarctic tundra ecosystem. The modelled warming responses using the adjusted T curve with a stronger BVOC T response showed good agreements with the observations, especially for the days with the observed strong emission responses to warming. Short-term underestimations were most likely linked to the underestimated leaf T

during the daytime. In the long-term (days-years), a mismatch in the modelled vegetation composition could also bring uncertainty in the simulation of emission responses to warming. The model estimated the mean annual isoprene and monoterpene emissions to be 20 and 8 mg C $m^{-2}$ $yr^{-1}$, with around 55 % and 57 % increase in annual emissions in response to a 2 °C warming for the period 1999-2012. For the warmest year, the 2 °C warming during the growing season resulted in 99 % and 94 % increase of isoprene and monoterpene emissions. These strong warming responses of arctic BVOC emissions have

hitherto not been specifically described in large-scale models and are therefore suggested to be included, especially in estimating regional emissions from the pan-Arctic.

**Author contribution**

J. Tang, G. Schurgers and R. Rinnan designed this research project. J. Tang did simulation runs, model developments and comparisons with the observation. G. Schurgers largely contributed to the research questions, model processes development

and calibration. R. Rinnan contributed to the research questions, data collection and project financial supports. H. Valolahti, P. Faubert, P. Tiiva and A. Michelsen provided field data used in this study. J. Tang wrote the manuscript and all authors critically read, commented, corrected and finally approved the manuscript.

**Acknowledgements**



This work was financed by the Villum foundation (VKR022589) and The Danish National Research Foundation (CENPERM DNRF100). The Danish Council for Independent Research | Natural Sciences, Emil Aaltonen Foundation, Maj and Tor Nessling Foundation, an EU ATANS grant (Fp6 506004), and the Abisko Scientific Research Station supported the field work providing the Abisko datasets. We would like to thank Frida Lindwall, Magnus Kramshøj and Ylva Persson for commenting on an earlier version of this paper. The authors thank Michelle Schollert and Ida Vedel-Petersen for providing leaf-level observational data. J. Tang would like to thank Paul A. Miller for interesting discussions about arctic PFTs.

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



## Tables and figures

**Table 1 Plant functional types (PFTs) and representative species in the study area. The emission capacity of isoprene ($I_S$, µg C gdw$^{-1}$ h$^{-1}$) and monoterpenes ($M_S$, µg C gdw$^{-1}$ h$^{-1}$) at 20 °C (in bold and italics) using the adjusted temperature response curve are presented as $I_{S20}$, whilst the averaged literature values based on the Guenther's algorithms with 30 °C as the standard temperature. The values are based on the available growing season leaf-level measurements from the Arctic.**

| PFT | $I_{S30}$ | $I_{S20}$ | $M_{S30}$ | $M_{S20}$ | Representative species names |
|---|---|---|---|---|---|
| Low Shrubs Evergreen (LSE) | 1.751 | *1.737* | 0.089 | *0.088* | *Empetrum hermaphroditum; Juniperus communis; Vaccinium vitis-idaea* |
| *Salix*, Low Shrubs Summergreen (SLSS) | 11.305 | *11.213* | 0.300 | *0.297* | *Salix phylicifolia; Salix glauca; Salix hastata; Salix myrsinites* |
| Non-*Salix*, Low Shrubs Summergreen (NSLSS) | 2.512 | *2.492* | 1.208 | *1.199* | *Vaccinium uliginosum; Betula nana* |
| Evergreen Prostrate Dwarf Shrubs (EPDS) | 1.411 | *1.400* | 1.312 | *1.301* | *Vaccinium oxycoccus; Cassiope tetragona; Dryas octopetala; Saxifraga oppositifolia; Andromeda polifolia* |
| Summergreen Prostrate Dwarf Shrubs (SPDS) | 14.117 | *14.003* | 0.428 | *0.425* | *Salix arctica, Arctostaphylos alpinus, Salix reticulata* |
| Graminoid Tundra (GRT) | 9.898 | *9.818* | 0.000 | *0.000* | *Calamagrostis lapponica, Carex parallela, Carex rupestris, Carex vaginata, Eriophorum vaginatum, Festuca ovina, Poa alpigena* |
| Cushion forbs, Lichens and Moss tundra (CLM) | 1.198 | *1.188* | 0.030 | *0.029* | *Astragalus alpinus, Astragalus frigidus, Bartsia alpina, Cerastium alpinum, Charmorchis alpina, Gymnadenia conopsea, Leucorchis albida, Pedicularis lapponica, Pinguicula vulgaris, Bistorta vivipara, Rubus chamaemorus, Saussurea alpina, Silena acaulis, Tofieldia pusilla, Hylocomium splendens Tomentypnum nitens, Pleurozium schreberi, Sphagnum warnstorfii, Peltigera aphtosa, Cetraria nivalis, Cladonia spp.* |





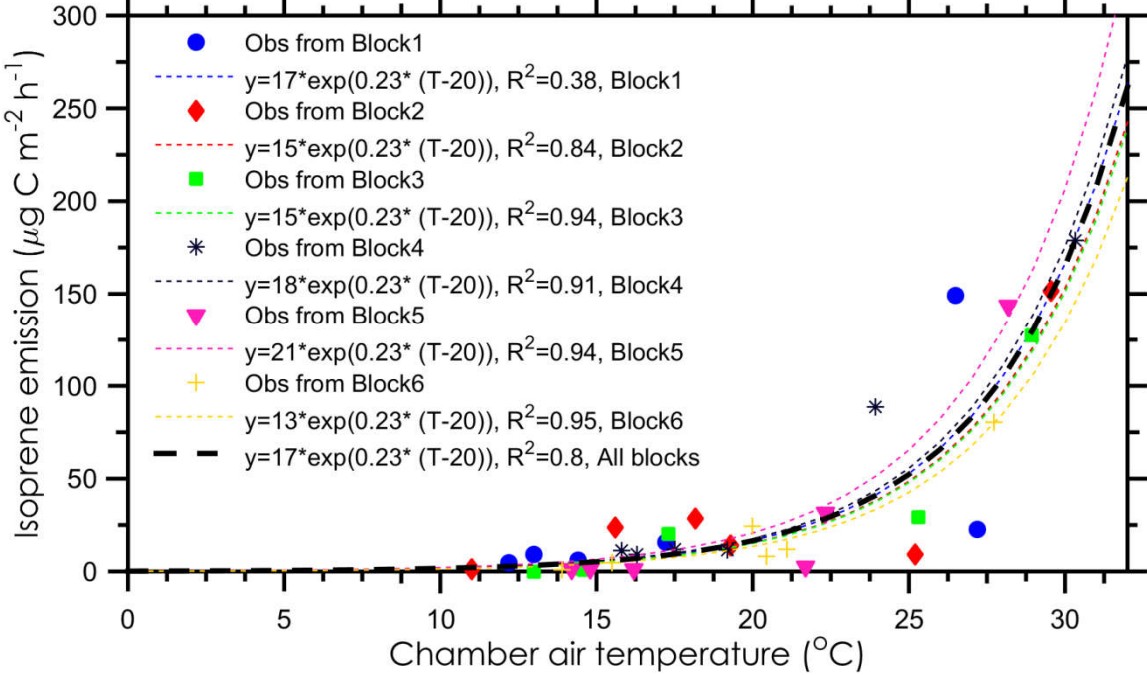

**Figure 1 The observed isoprene emission rates in relation to the chamber air temperature in July over three field campaign seasons (2006, 2007, 2012) in the Abisko heath tundra.**





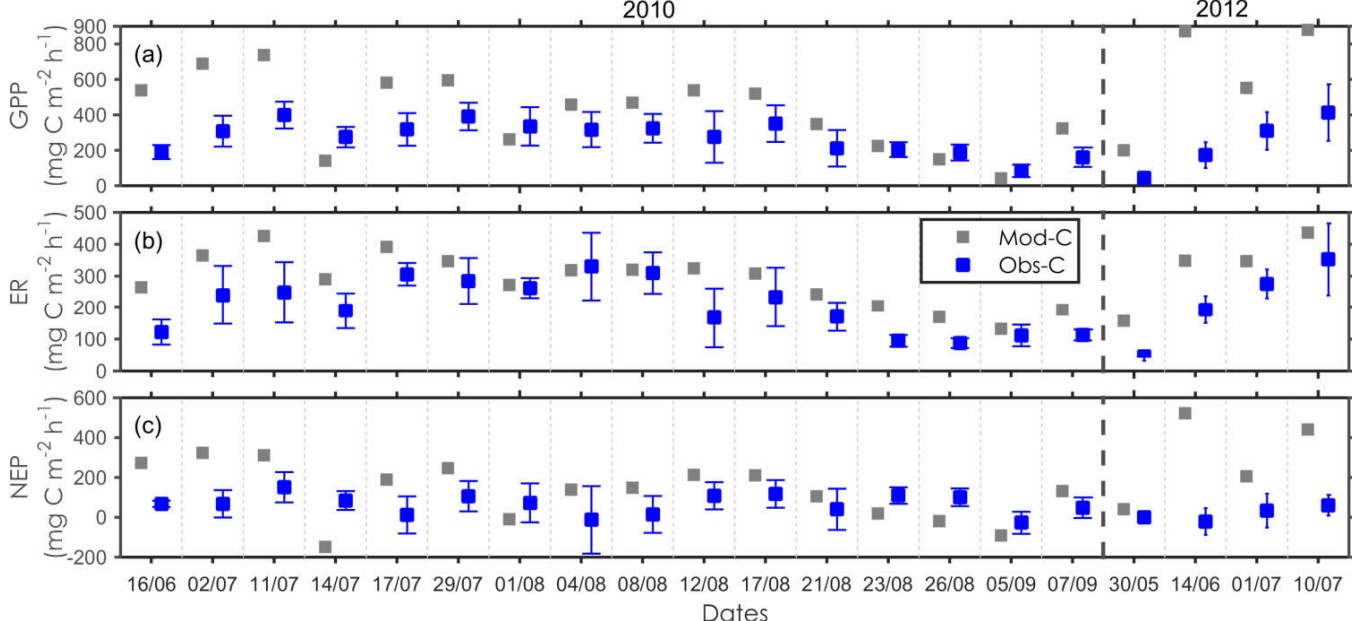

**Figure 2 Modelled (grey) and observed (blue) gross primary production (GPP, (a)), ecosystem respiration (ER, (b)), and net ecosystem production (NEP, (c)) for the growing season of 2010 and 2012 in the control plots at the Abisko heath tundra. Error bars indicate the standard deviation for the six replicates**





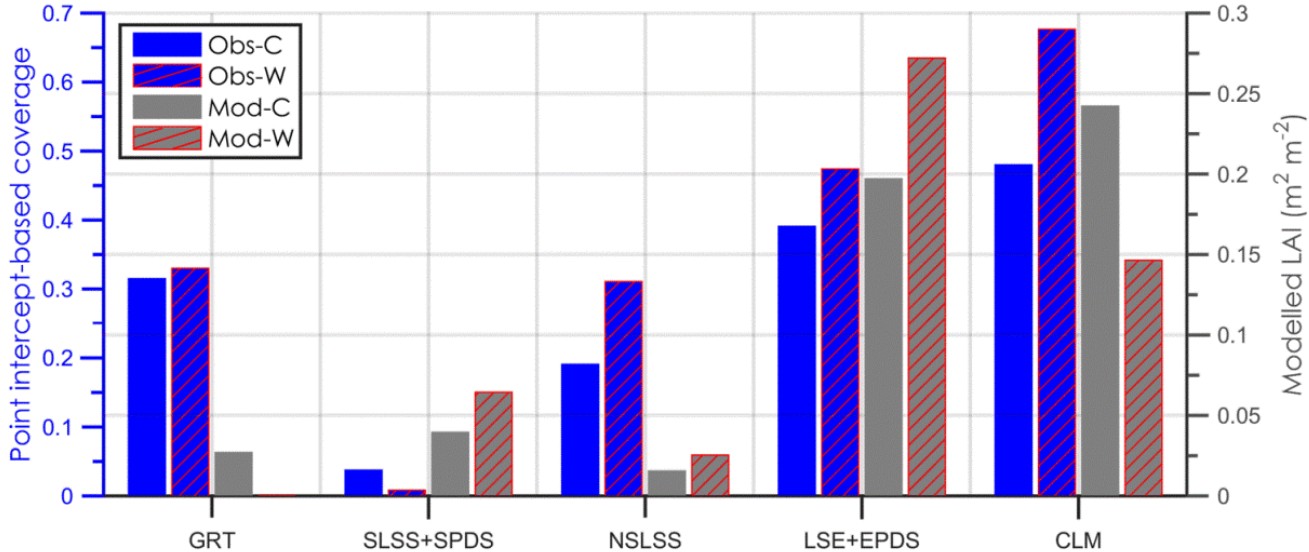

**Figure 3 Point-intercept based vegetation coverage (during the growing season) and modelled growing season leaf area index (LAI, $m^2\ m^{-2}$) averaged for the year 2010 and 2012 for the control (C) and warming (W) treatments in the Abisko heath tundra. Different y axes are used for the observed (Obs) and the modelled (Mod) coverage. GRT: Graminoid tundra; SLSS: *Salix*, low shrubs summergreen; SPDS: Summergreen prostrate dwarf shrubs; NSLSS: Non-*Salix*, low shrubs summergreen; LSE: Low shrubs evergreen; EPDS: Evergreen prostrate dwarf shrubs; CLM: Cushion forbs, lichens and moss tundra.**





**Figure 4 Time-series of the air temperature (Air T) at 2 m height, photosynthetically active radiation (PAR), the modelled isoprene (ISO) and monoterpene emissions (MT) (µg C m⁻² h⁻¹) for the days 150-250 in 2006, 2007, 2010 and 2012 in the Abisko heath tundra. Both modelled and observed fluxes are from the control (C) conditions and the modelled daily average (Mod-C) and daily noon (Mod-C: noon) emissions are presented. Error bars indicate the standard deviation for the six replicates. For the year 2010, isoprene emission rates were not analyzed due to technical problems.**

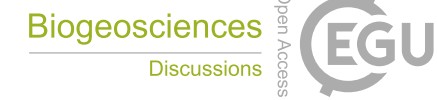



**Figure 5 Comparison of the modelled daily average and noon isoprene emission rates with the observations in the control (C) plots (a) and evaluation of modelled warming responses (WR) with the observed WR (b) at the Abisko heath tundra. The modelled daily average (daily leafT) and noon leaf temperature (noon leafT) in the C plots were compared with the observed chamber air temperature (airT) averaging over sampling times during daytime. Mod: Modelled; Obs: Observed.**





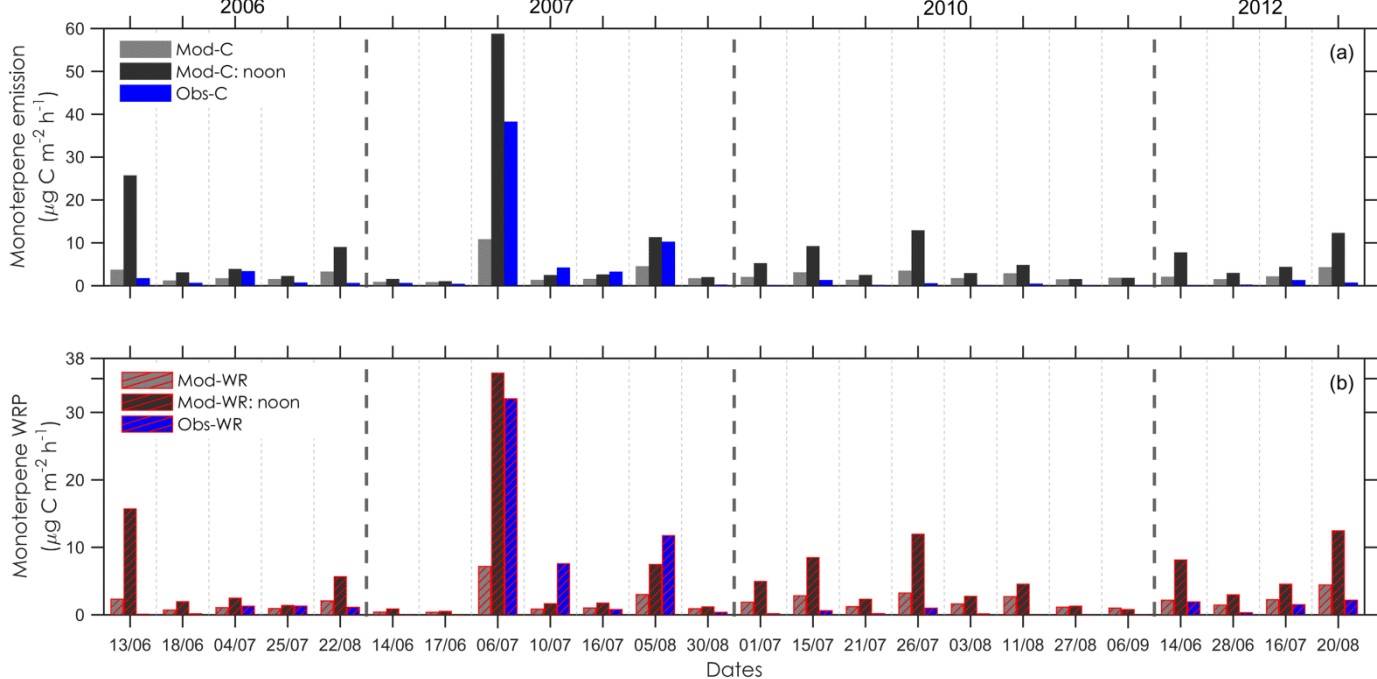

**Figure 6 Comparison of the modelled daily average and noon monoterpene emission rates in the control (C) plots (a) and the evaluation of modelled warming responses (WR) with the observed WR (b) at the Abisko heath tundra. Mod: Modelled; Obs: Observed.**





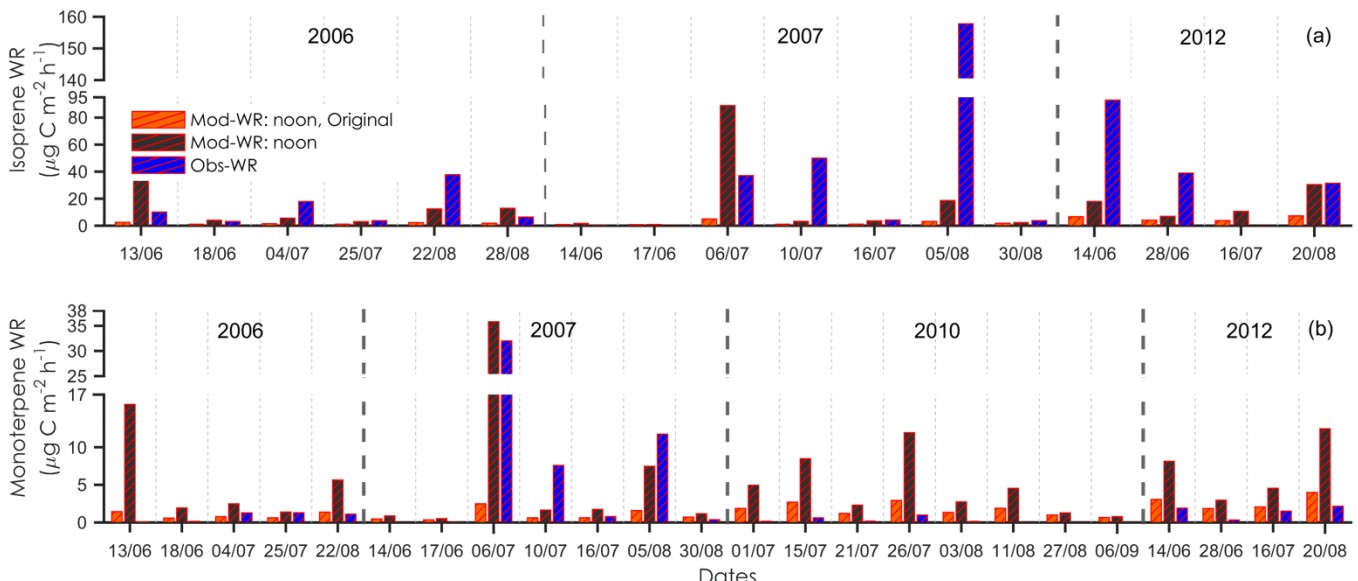

**Figure 7 Comparison of the modelled daily noon isoprene and monoterpene warming responses (WR) from the simulation with the original temperature responses (Mod-WR: noon, Original) and the simulation with the adjusted temperature responses (Mod-WR: noon). The observed WR (Obs-WR) are also presented.**





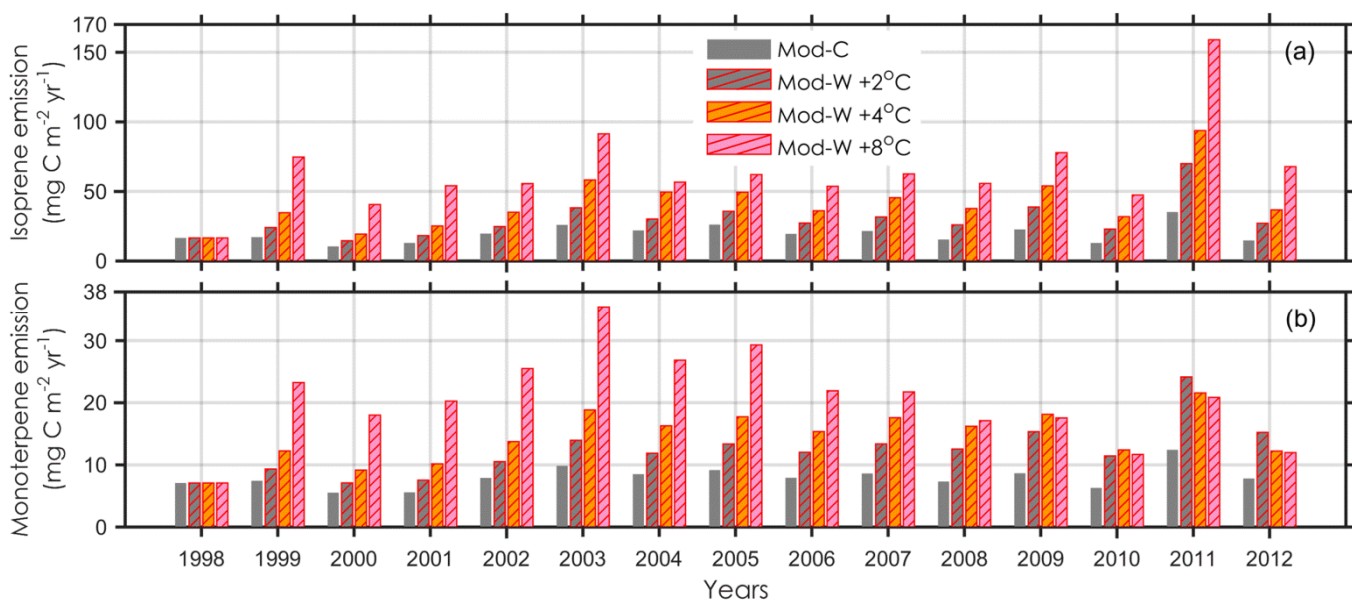

**Figure 8 Modelled annual isoprene and monoterpene emissions for the period 1998-2012 at the Abisko heath tundra. The warming (W) treatment started in 1999 and three levels of warming (+2 °C, +4 °C and +8 °C) were applied during summertime. The modelled annual emissions in the control (C) plots are also presented.**

