# Peer review of "Challenges in modelling isoprene and monoterpene emission dynamics of arctic plants: a case study from a subarctic tundra heath"

_Biogeosciences, 2016_

## Referee Comment (RC1) · Anonymous Referee #1 · 9 May 2016

Major comments

The subject matter of this paper is important. The Arctic environment is changing rapidly. Because of BVOC impacts on air chemistry, it's important to have models that can successfully predict the response of BVOC emissions. This paper makes an important contribution by employing a model with a dynamic vegetation component. As they warm, Arctic ecosystems are expected to see a shift towards woody plants, and this should change the capacity of the ecosystems to emit BVOCs.

The paper has strengths, but also needs substantial improvements before publication. The basic modelling approach is sound, and it's helpful that the authors include the investigators that actually made the measurements. The paper demonstrates a good understanding of many of the ecosystem processes that should be captured by the

model. Overall, I thought the discussion section was strong. Among weaknesses, the comparison of the model to the observations need to be improved. First, much of the discussion is qualitative. The model is said to fit the observations well in many instances, but there is no quantitative analyses: no goodness of fit metrics, and no statistics. Need to formally compare model to observations with statistics. More specifically, using the max and the daily average as a basis for comparison doesn't make much sense. What is the point of a daily average, especially since the meaning of the daily average changes with the long diurnal cycles in the Arctic? Why not just use the times of day that cover the range of the observations? Also, the figures could be improved by consolidation. The same data are presented in multiple figures in two different instances. The figures would also be easier to interpret if instead of presenting the max/daily average, just one metric was used for comparison to the observations. Also, there is very little acknowledgement of potential for experimental error in observations (one mention at the very end). Given the technical challenges with experiments in the Arctic, the potential for measurement error should be addressed.

The employed model is touted as being a mechanistic model, but then an empirical method is used for its calibration to the dataset. This is not itself a problem per se, but the paper states that mechanistic models are better than empirical models. If so, why is such an empirical calibration necessary? Also, a serious deficiency with the model is that it does not account for the effect of previous weather conditions (example, 24 hours and 10 days) on the capacity to emit BVOCs. This effect is potentially very important in the Arctic.

Finally, the list below of minor comments and technical corrections is extensive.

Minor comments and technical corrections

Title: The title suggestions that the article will focus generally on modelling subarctic plants, but instead the article is about one specific effort using one specific model formulation. While of course some of the manuscript is more general, it is also uses

data from just one field site.

Page 1, line 14 – page 2, line 4: The abstract could be clearer. There are some specific recommendations below, but more generally the abstract should be condensed and just the highlights presented.

Page 1, line 14: Title says "subarctic" while abstract goes back and forth between arctic and subarctic. Make sure each use is intentional. Further in the manuscript, (sub)arctic is used. Again, make sure this is all consistent.

Page 1, line 23: "higher levels of warming" instead of "higher levels' warming".

Page 1, lines 24-26: The sentence should be written. Do you mean the "measured" BVOC WR, not modeled? If you do mean modeled, what was the standard "were better captured"? Also, "compared" instead of "comparing".

Page 1, line 26: This sentence is relays an interesting result, but there is not enough context to warrant inclusion in the abstract. Please remove it.

Page 1, lines 30-31: This sentence can be removed, since it's a circular argument. The high WR led to the high adjustment T curve.

Page 2, line 3: remove "extrapolation".

Page 2, lines 3-4: How do points (2) and (3) differ? Isn't "PTF's responses to warming" a subset of "representation of vegetation dynamics in the past and future"?

Page 2, line 7: "plant" instead of "plants" or include an apostrophe.

Page 2, lines 10-13: First, need to include that BVOCs don't solely react with OH. In particular, ozone is another important reaction partner for some BVOCs. Second, in a low-NOx environment, BVOC emissions can lead to a reduction in tropospheric ozone concentrations.

Page 3, line 3: "from" instead of "along". Also, why is G3P the "chief precursor" if

pyruvate is also required?

Page 3, line 6: "part of monoterpene productions" should be clarified.

Page 3, line 8: remove the inner set of parentheses.

Page 3, lines 13-15: This is a contentious statement, and there is no reference. A more nuanced statement is necessary, and could reference Monson, R.K., Grote, R., Niinemets, U., Schnitzler, J.P., 2012. Modeling the isoprene emission rate from leaves. New Phytologist 195, 541-559.

Page 3, line 16: "referred to here" instead of "referred here"

Page 3, line 17: remove space before comma

Page 3, lines 18-20: Should reference Potosnak et al 2013 here. While dwarf willow's T response was OK compared to G93, the light response was more linear than expected.

Page 3, line 25: Should also include low transpiration rates. Because of permafrost, transpiration rates can be low, which also leads to the high ground temperatures.

Page 3, line 30: Give what LPJ-GUESS stands for.

Page 4, lines 2-4: The objectives could be clarified. To me, (1) "capture the observed BVOC T sensitivity" is the same as part of (2) "To address short-term and long-term impacts of warming on ecosystem BVOC emissions." Be more specific about your study goals, or further differentiate the difference between 1 and 2.

Page 4, line 8: use straight single quote for minutes symbol.

Page 5, line 1: You have already defined PFTs above, so don't redefine.

Page 5, lines 2-4: Is this statement true for Arctic-specific PFTs? Please indicate this.

Page 5, line 5: What does "large-scale" mean here? I consider the base Farquhar equations to be leaf-level. Do you mean canopy-scale?

Page 5, lines 4-12: I assume transpiration & stomatal conductance are also modelled to get pi? Maybe you'll talk about this further down, but it would be important for understanding discrepancies between air and leaf temperature.

Page 5, line 18: How is Ci different from pi defined on line 11. Just concentration vs. partial pressure? What does "without water stress" really mean? This is probably tied to my comment above.

Page 5, line 26: "optimum from terpenoid synthesis" should be "optimum for terpenoid synthesis"

Page 6, lines 1-2: Give a reference for the co2 response in the model, as you've done for the other responses.

Page 6, lines 12-15: Again, this gets back to my comments above about transpiration and conductance. It would make more sense to move this discussion to the general description of the model, before discussing biogenics. Also, more detail on this part is necessary. What are the details here? This can be done by references to the literature, if it has been described by LPJ-GUESS before. What is the coupling between estimating leaf temp, internal co2, transpiration and stomatal conductance? Or is a more empirical algorithm used?

Page 7, line 4: Fix grammar: either "appearing" or change sentence structure.

Page 7, line 4-5: I agree there is insufficient data, but mosses may make a large contribution to BVOC emissions in some Arctic ecosystems. So, it's fine to incorporate them into a larger PFT, but are you capturing their emissions? That is, do the emission factors for this PFT reflex the mosses?

Page 7, line 17: first, not firstly.

Page 7, line 20: "other" instead of "rest"

Page 7, lines 18-21: Given the lack of data for the Arctic, it's justifiable to use two years

data for calibration and two years for validation. But, the sensitivity of this procedure should be assessed by flopping the years: how different are the results if the second two years are used for calibration, and the first two used for validation?

Page 8, lines 9-11: The goodness of fit here is a bit deceptive. The fit is entirely driven by the relatively few points that are above 23 deg C. Since everything below that is relatively close to zero, there is little new information added. For example, blocks 5 and 6 only have one observation each above 23 deg C, so the individual fits are very good. I don't see the added value in the doing the individual fits for each block. It seems all that info comes out of the overall fit. Finally, you should understand the justification for using 20 instead of 30. Yes, this makes sense conceptually and certainly for measurements, but realize that mathematically, using your formulation, there is no difference between using 20 and 30, because of the laws of exponents. That is, you'll get the same r2 for the fits with each. This isn't true with more complicated formulations of the T response; for example, the T response in isoprene emission for G93.

Page 8, lines 16-23: This is confusing. Your goal is to compare your measurements to the model. So, yes, using daily averages isn't appropriate. But why discuss them in the first place? I think you'll use them for another purpose, but that's not clear. Why do you use max T & PAR? Wouldn't an average around the measurement time make more sense? And again, your last sentence here is obvious. Particularly in the Arctic, with low sun angles for much of the day, this isn't a strong statement.

Page 8, lines 27-28: Again, examine Equation 3. You'll see that changing from 30 to 20 only introduces a constant.

Page 9, line 4: In Fig. S1, the figure legend should indicate what the dashed vertical line denotes at the value of 0.4 in both panels. The text explains this, but the figure caption should too.

Page 9, line 10: Do you expect to see a one-to-one correspondence between the point intercept info and the LAI values? This surely doesn't hold as LAI gets closer to 1 (and

exceeds it), but you should share your expectation here. Do you assume that there is no overlap with cover, and therefore there should be a one-to-one relationship? If so, state that.

Page 9, lines 18-22: You discussed the LAI response to warming, but not the GPP/NEP/ER response. Why?

Page 9, line 29 – page 10, line 1: This analysis isn't adding much to your argument. Of course you see this, because your model is driven by PAR and T. You don't need to cover this result—it follows directly from your model formation (Equations 1-3). Second, I don't understand the relevance of relating mean daily ISO/MT production to the noontime values. What do you learn from this?

Page 10, lines 3-4: For "the observed average rates (blue squares) were well captured by the modelled noon emissions" you need to present some statistics to back up this statement. You should do an xy plot of this data and see what the fit looks like. Even if you don't present the plot as a figure, you should report the statistics of the fit.

Page 10, line 10: As mentioned below, the same data is presented in Figs 4 and 5a. And now you've made the same statement about fit as above. This should be consolidated, and again there needs to be a statistical analysis of the goodness of fit.

Page 10, lines 13-18: Yes, the temperature drives these emissions, but this is a bit complicated because of the chamber observations. There are two issues: one, the model's ability to predict leaf T; second, the increase in air T because of the chamber used to measure BVOCs. Only the first is important for extrapolating your results.

Page 10, lines 25-27: Again, need statistics to back up these contentions.

Page 11, line 30: After not using any statistics comparing the model to observations, why would you use a statistic in this case, when you are comparing the model to itself?

Page 13, lines 23-25: This is an interesting contention. But, the emissions for the storage pool are generally regarded as being due to the physical process of evaporation

of the MTs. Why would this change for Arctic plants?

Page 13, lines 28-29: Yes, and that is why restricting your modelling to the times of day when measurements occurred would help.

Page 14, lines 2-3: Do you mean the bryophyte decrease due to drying is an artifact of the experimental warming and shouldn't be captured by the model? Please elaborate.

Page 14, lines 5-7: Yes, because you used the observed data to fit your model. Remind readers of that point.

Page 14, lines 11-13: Could also mention the drying that was noted above for species responses (bryophytes).

Page 14, lines 18-19: Need some more analysis here. Yes, the two responses are very important. But, you should re-emphasize that your dynamic vegetation model isn't doing a great job of getting the vegetation changes correct. Therefore, the results in Fig 8 are illustrative of the impact, but the details are not certain.

Page 14, lines 25-30: I agree with most of this logic, but since this particular study is looking at whole system measurements, there is potentially an interaction between the true T response of the plants and the issue of canopy temperature described earlier. You should at least discuss the possibility that some of this T response is not at the enzymatic level, as suggested here, but is due to a non-linear increase in leaf T with increasing air T due to canopy warming. Perhaps some of the references cited are leaf-level measurements which could clarify this point?

Page 15, lines 1-3: Yes, this is important, but it also brings in the issue of drought stress. Drought stress can occur frequently in some Arctic ecosystems due to relatively shallow soils above the permafrost. To understand canopy heating, it will be necessary to understand canopy water dynamics.

Page 15, lines 15-16: Great this is stated clearly in the conclusion, but this point should also be made in the discussion.

Figures 2, 4, 5 and 6: For Figs 2, 5 and 6, use you DD/MM on the time axis, but day of year for Fig 4. Be consistent, and I prefer day of year.

Figures 5, 6: The top panels (a) of each figure are the same data presented in Figure 4. These results shouldn't be presented twice.

Figure 7: There is also a lot overlap with Figures 5b and 6b: two of the three sets of data have already been shown. In addition, why is there a break in the y-axis, when mostly the same data have been presented in Figures 5b and 6b without a break?

Figure 9: Why include the higher-T scenarios? I understand they are (unfortunately) realistic due to the IPCC estimates of climate change. But, you don't discuss them much, and there are obviously some weird things happening with the vegetation change (for example, lower +8 compared to +2 for MTs in 2012). Since the vegetation changes predicted by the model are suspect, the results of the +4 and +8 runs are highly speculative.

---

## Referee Comment (RC2) · Anonymous Referee #2 · 10 May 2016

General comments:

This is a nicely written manuscript which addresses an important question in BVOC estimation – namely the representation of cold environments in global estimates and the uncertainties of modelling in this respect. It is also well timed since a lot of new information has recently been published about this topic and the implementation of this knowledge into a model is overdue. However, I feel like I have to urge the authors to be more careful in what they regard as 'good agreement' between measurement and simulation or at which point they conclude that the model's suitability has been 'demonstrated'. Overall, I see a lot of model deficiencies and uncertainties in this study which should probably be the prime focus of the investigation. In this respect, I would welcome figures or statistics that show the actual relation between measurements and simulations rather than column- or point diagrams. Apart from this, I think that the

model description part needs some elaboration.

Specific comments:

P1, L22: 'Short time scales' not only need to be defined, mentioning them here is also irritating. In fact, the question about simulations and observations referring to different time periods is troubling me throughout the manuscript.

P1, L24: The model 'was able' to reproduce carbon fluxes for the majority of the vegetation period but showed considerable weakness in representing the seasonality, probably due to mismatch of phenological phases. This should be recognized.

P1, L26: The difference of effective temperature in model and observation is certainly one reason for a mismatch in emission simulations which has been correctly acknowledged here. However, giving this as the only reason for a possible deviation is misleading at this point.

P2, L17ff: Major uncertainties are also other driving factors for emissions that are usually not considered in models, namely air chemistry, soil water availability, UV light and biological stress impacts. Also the representation of seasonality (which is composed of phenology and enzymatic activity changes) is a point worth mentioning here. The authors are mentioning most of these points at a later stage but I feel that it needs mentioning here.

P3, L5: I think that in the Pacifico and Unger papers, the Niinemets approach is used. So this is to some degree a repetition here.

P3, L10: seasonality and/or past weather conditions? In fact this is the same problem. You might differentiate into effects of phenology and enzymatic activity shifts though.

P4, L15: From the later remarks I take it that the BVOC emissions were not taken round the clock so the time or time period during the day when the measurements were made should be mentioned.

P5, L7ff: I am a bit irritated here. The Haxeltine and Prentice photosynthesis approach is for seasonal or annual photosynthesis estimation, assuming a kind of optimal adjustment to average environmental conditions. Nevertheless, the model seems to work on daily timesteps here. The description given about the model itself looks very much like the Collatz approach – so what is taken from Haxeltine here? Regarding the description, many abbreviations are introduced here that seem not to be used later on – please check.

P5, L14ff: Since emissions depend on temperature in a highly non-linear fashion, I think it is generally acknowledged that calculating them with daily average values is necessarily not capturing the dynamics. Regarding the Niinemets model, for example Unger et al. used a 15 minutes time steps. From the description it sounds like LPJ feeds daily photosynthesis results into daily emissions. Can you elaborate on the problem? Also, I think that the reference temperature used in equation 3 and/or the parameter in the response function needs to be adjusted because the model is not using them as an immediate response value anymore but as parameter for daily average emission. (30 degrees as an average value throughout the day would probably exhaust the emission apparatus so that the response curve would not be valid anyway.)

P5, L15: Instead of using I for isoprene as well as monoterpenes shouldn't you use Ei and Em or similar? This can further be modified for storage (e.g. Ems) in equation 4.

P5, L22: Here, the influence is named 'phenology' while later the same function refers to 'seasonality' (L30). Since these are two different things – is this a lumped index? Specific or specifically parameterized for PFTs? Empirical or dependent on weather or climate?

P5, L27ff: see also comment from L14ff. It seems that the reduction of reference temperature is rather a necessity from applying the model on a daily time step than a particular feature of arctic plants.

P5, L29: it is stated that the reference temperature is changed. This is to 20 oC as

elaborated on later, correct?

P6, L2: fCO2 according to? Since it seems that variable CO2 air concentrations are used, it would be helpful to know to which degree CO2 might be responsible for differences between the years (probably small, but anyhow. . .).

P6, L14: If the energy balance calculation was modified specifically for this study and is not published elsewhere, this modification should be explained.

P8, L15: I don't get how this can give you LAI values. Could you elaborate a bit? Looking at figure 3 there seems to be a difference between Lai and what is measured but the measurements are nevertheless used for evaluation. So how are the two related?

P8, L21: I agree that model results in daily resolution might not be comparable to measurements done at noon. This seems to be a general problem as mentioned above. I also agree that you can calculate noon temperature from average temperature to get a representative value of noon emission – but why don't you do the same with PAR? Instead of using the average value which is definitely wrong you can estimate maximum PAR from average PAR (e.g. Berninger F (1994) Simulated irradiance and temperature estimates as a possible source of bias in the simulation of photosynthesis. Agric. Forest Meteorol. 71:19-32)? Have you estimated the sensitivity of this error on the results?

P9, L3: Check wording. I think it should be the modelled co2 fluxes that are sensitive to a change of parameter. This should also be indicated in some kind of measure, i.e. the degree to which the parameter was varied.

P9, L9ff: In fact, the deviations are considerable. Not only GPP and thus emission is considerably overestimated in both years early seasons – which should be quantified and considered in annual estimates – but LAI is totally wrong in all PFTs except LSE+EPDS and CLM under current climate where the overestimation is a mere 10-15 percent. In L15/16 it is stated that these are the most important PFTs but in the

next sentence the other PFTs are described to have a 'large coverage'. Are there any numbers that I have missed that give an objective picture about the abundances?

P10, L5: Monoterpene emissions seem to be met particularly because measurements occurred mostly on days with low emissions (according to figure 4). This is a problem because the high simulated emissions practically lack evaluation that should be addressed. I can certainly imagine other ways of representation or statistical analysis that can be used to elaborate on the point.

P10, L10: Similarly, I have large difficulties agreeing that figure 5 supports the statement that isoprene emissions were mostly captured by the model.

P11, L26ff: The simulated annual emissions include the largely wrong response of LAI as well as the wrong response in early season emission, right? Can the error somehow be estimated? I have the feeling that these calculations might be too far off to be considered here.

P12, L14ff: The discussion seems to be overall comprehensive. Still, as for example in the first line, I think the authors are overenthusiastic about their results. This also applies for the conclusions.

P14, L23ff: The comparison with common parameterization should not only be concentrated on the arctic environment but also on the problem with the time resolution (see above).

---

## Referee Comment (RC3) · Anonymous Referee #3 · 13 May 2016

Overview:

This paper presents a very valuable and interesting work, focusing on isoprene and monoterpene emissions from subarctic plants, a topic that has not been investigated or published much so far. I really appreciate the originality of this study, which helps to improve our understanding regarding emission estimates. However, as also raised by the two other referees, I think that this manuscript would really benefit from a deeper and more detailed presentation, of the result analysis and discussion especially, which would help to appreciate more clearly the validity of the conclusions of this work. Here are some feedbacks and corrections that would need to be considered before publication in BGD, that I warmly support.

General comments:

Abstract: in "evaluating BVOC related processes", which processes for instance do you refer to, photosynthesis?

Generally in the manuscript, the analysis is rather qualitative than quantitative and should be more detailed and specified. Some elements giving more precise information on the context could also be added. For instance, what is the estimated contribution of subarctic plants to global isoprene and monoterpene emissions? This could be specified for both the present-day case and the different warming scenarios, giving more perspective to the work carried out, and is important to be discussed, especially in section 4.

Page 5, section 2.2.2 BVOC modelling: Could you please detail what the seasonality function used in isoprene production calculation stands for?

Page 6, section 2.2.2 BVOC modelling: Works published so far agree on the $CO_2$ inhibition effect regarding isoprene emissions, but not regarding other BVOC emissions. Is the $f(CO_2)$ function considered in the model only for isoprene or for every BVOCs? On which work is it based and is the same parameterization considered for every compounds?

Page 9, line 25: What do you mean exactly with "dynamic vegetation" in "simulating dynamic vegetation enables us to assess the model performance"? Day-to-day variability? Higher frequency? Indeed the term of dynamic vegetation can also refer in vegetation modeling to long-term changes in vegetation distribution due to climate and $CO_2$ changes.

The model/data comparison would also really benefit from a deeper analysis. If isoprene and monoterpene emission estimates fall into the data values, it is however difficult to come to a clear conclusion, as data are not that numerous, and as model estimates are given either as daily average or for noon. At what time were emission data collected and how are they compiled for model-data comparison?

The parameterization is calibrated and adjusted in order to better represent BVOC emission from Arctic plants. This is a crucial and one major contribution of this work and yet it is only very quickly mentioned in section 4.2. It is important to add a more detailed and quantitative analysis of the emission improvement, both in the results section and in the discussion part.

Specific comments:

Page 1, line 23: change "the model's responses" to "the model responses"

Page 1, line 23: change "higher levels' warming" to "higher warming levels"

Page 2, line 4: change "ÂÍPFT's reponses" to "PFT responses"

Page 2, line 6: change "Biogenic volatile organic compounds (BVOC)" to "Biogenic volatile organic compounds (BVOCs)"

Page 2, line 11: change "atmosphere's oxidative capacity" to "atmosphere oxidative capacity"

Page 2, line 15: change "(. . . respectively (Sindelarova et al., 2014))" to "(. . . respectively; Sindelarova et al., 2014)"

Page 3; line 29: change "and to advance our understandings of the" to "and our understanding regarding"

Page 3, line 30: remove comma in "ecosystem model, LPJ-GUESS"

Page 7, line 28: please change "LAI of the year 2006 and 2007" to "LAI of the years 2006 and 2007"

Page 7, line 33, change "due to plants' adaptation" to "due to plant adaptation"

Page 8, line 20-21: change "Due to lacking of data about the daily maximum" to "Due to the lack of data regarding the daily maximum"

Page 9, line 26: change "to assess the model's performance" to "to assess the model

performance"

Page 15, line 11: change "the model's ability" to "the model ability"

Figures:

It is hard to distinguish the observations from the emission estimates. Could you please trying using another color?
* * *

---

## Author Comment (AC1) · 23 Jun 2016

Thanks for the reviewer's constructive comments and suggestions for improving this manuscript. All comments have been answered. Below, we first give general answers to the reviewer's comments and the detailed comments will be addressed separately point by point.

Major comments

The subject matter of this paper is important. The Arctic environment is changing rapidly. Because of BVOC impacts on air chemistry, it's important to have models that can successfully predict the response of BVOC emissions. This paper makes an important contribution by employing a model with a dynamic vegetation component. As they warm, Arctic ecosystems are expected to see a shift towards woody plants,

and this should change the capacity of the ecosystems to emit BVOCs. The paper has strengths, but also needs substantial improvements before publication. The basic modelling approach is sound, and it's helpful that the authors include the investigators that actually made the measurements. The paper demonstrates a good understanding of many of the ecosystem processes that should be captured by the model. Overall, I thought the discussion section was strong. Among weaknesses, the comparison of the model to the observations need to be improved. First, much of the discussion is qualitative. The model is said to fit the observations well in many instances, but there is no quantitative analyses: no goodness of fit metrics, and no statistics. Need to formally compare model to observations with statistics. More specifically, using the max and the daily average as a basis for comparison doesn't make much sense. What is the point of a daily average, especially since the meaning of the daily average changes with the long diurnal cycles in the Arctic? Why not just use the times of day that cover the range of the observations? Also, the figures could be improved by consolidation. The same data are presented in multiple figures in two different instances. The figures would also be easier to interpret if instead of presenting the max/daily average, just one metric was used for comparison to the observations. Also, there is very little acknowledgement of potential for experimental error in observations (one mention at the very end). Given the technical challenges with experiments in the Arctic, the potential for measurement error should be addressed.

The employed model is touted as being a mechanistic model, but then an empirical method is used for its calibration to the dataset. This is not itself a problem per se, but the paper states that mechanistic models are better than empirical models. If so, why is such an empirical calibration necessary? Also, a serious deficiency with the model is that it does not account for the effect of previous weather conditions (example, 24 hours and 10 days) on the capacity to emit BVOCs. This effect is potentially very important in the Arctic.

Finally, the list below of minor comments and technical corrections is extensive.

Response: (1) We agree with the reviewer that the comparison of modelled and observed variables should include some statistics and will add Willmott's index of agreement (describing models' prediction with pairwise-matched observations) as well as mean bias error (describing mean deviations between modelled and observed values) for each comparison.

(2) The reason why we present both daytime average and daily maximum values to compare with the observed is that the BVOC sampling on each field plot was conducted at a certain time point of a day for 30 min, while the modelled processes are at daily scale. Considering the strong diurnal cycle of BVOC emissions (for an arctic example, please see Lindwall et al. (2015)), neither daily average nor daily maximum were accurate enough to directly compare with the observations. We reason that by presenting both rates, we can see the range of the modelled daily emissions.

(3) There is some data repetition in Figures 4-6, and separating them in different figures was aiming to explain different perspectives. In the revised manuscript, we will move Figure 4 to the supplementary to reduce data repetition. There will be no data repetition in Figure 5 and 6.

(4) We agree that the discussion about potential uncertainties from the measurements should be addressed in depth, mainly covering point intercept-based coverage and side effects from OTC-chambers.

(5) Regarding to the empirical method used in calibrating T response, we agree with the reviewer that there is empirical element in the parameter estimation, which, however, reflects some processes understanding, e.g. underlying enzyme activation. Even for mechanistic models in bio/geosciences fields, it sometimes cannot avoid of using empirical relationships determined from observations where multiple processes may anticipate. In our case, the calibrated T response (empirically) is used for influencing fraction of photosynthetic electron transport contributing to isoprene and monoterpene production, which is internally linked to other processes and potentially reflect more

dynamics than other empirical models.

(6) The current model did not consider the past weather conditions, rather, it emphasizes the enzymatic acclimation to short-term climate. As we do not have observed data to support which period of past weather should be taken into consideration, adding emission acclimation to the past weather will bring additional uncertainties to the modelled fluxes and complicate the current comparison. Ekberg et al. (2009) has fitted their observational data to obtain a relationship between past weather conditions (48 h) and isoprene emissions (not for monoterpene) from wetland sedges. We would like to further address this issue in the future model development with assistance of available climate data on measurement sites (past 24-96 h temperature as well as leaf level BVOC sampling data).

Minor comments and technical corrections

Title: The title suggestions that the article will focus generally on modelling subarctic plants, but instead the article is about one specific effort using one specific model formulation. While of course some of the manuscript is more general, it is also uses data from just one field site.

Response: We can well understand the reviewer's concern about including only one study site and will therefore add a sub-title to specify it. The title will be changed to "Challenges in modelling isoprene and monoterpene emission dynamics of subarctic plants: A case study from a tundra heath".

Page 1, line 14 – page 2, line 4: The abstract could be clearer. There are some specific recommendations below, but more generally the abstract should be condensed and just the highlights presented.

Response: Thanks for the comments. The abstract will be adjusted to condense the length and to make points clearer.

Page 1, line 14: Title says "subarctic" while abstract goes back and forth between arctic

and subarctic. Make sure each use is intentional. Further in the manuscript, (sub)arctic is used. Again, make sure this is all consistent.

Response: We will carefully consider each mention of "subarctic" and "arctic" through the whole manuscript. We will check and correct to use the most precise term in each place. The observational data originate at the Subarctic, but many ecosystem processes and components function similarly both in the Subarctic and the Arctic, and many of the issues handled are similar in both regions.

Page 1, line 23: "higher levels of warming" instead of "higher levels' warming".

Response: Accepted.

Page 1, lines 24-26: The sentence should be written. Do you mean the "measured" BVOC WR, not modeled? If you do mean modeled, what was the standard "were better captured"? Also, "compared" instead of "comparing".

Responses: Yes, it should be "measured". The suggested changes will be implemented.

Page 1, line 26: This sentence relays an interesting result, but there is not enough context to warrant inclusion in the abstract. Please remove it.

Response: The reason behind is the underestimated leaf T is one of the main factors influencing short-term BVOC fluxes. The sentence sounds a bit misleading, but we will clarify it.

Page 1, lines 30-31: This sentence can be removed, since it's a circular argument. The high WR led to the high adjustment T curve.

Response: The adjustment of T curve was only based on the emission rates from the control plots under naturally varying weather conditions within the growing season, and no emission rates from the warming plots were used. So, it is not high WR, which led to the high T curve. The improved ability of capturing the observed WR by the adjusted

T curve indicates a better representation for arctic plants.

Page 2, line 3: remove "extrapolation".

Response: Changed.

Page 2, lines 3-4: How do points (2) and (3) differ? Isn't "PTF's responses to warming" a subset of "representation of vegetation dynamics in the past and future"?

Response: we agree that these two points sound similar. Here, in point (2) "PFT's responses to warming", we mainly mean plant's physiological adaption to warmer climate, while in point (3) "representation of vegetation dynamics in the past and future", we mainly refer to long-term vegetation development, e.g., composition changes, disturbances, and expansion, etc. We will clarify the sentence.

Page 2, line 7: "plant" instead of "plants" or include an apostrophe.

Response: Accepted.

Page 2, lines 10-13: First, need to include that BVOCs don't solely react with OH. In particular, ozone is another important reaction partner for some BVOCs. Second, in a low-NOx environment, BVOC emissions can lead to a reduction in tropospheric ozone concentrations.

Response: The text will be clarified. Thanks for the points.

Page 3, line 3: "from" instead of "along". Also, why is G3P the "chief precursor" if pyruvate is also required?

Response: We have changed the words. The description of G3P as the chief precursor was not accurate and will be corrected.

Page 3, line 6: "part of monoterpene productions" should be clarified.

Response: This is not accurate. It will be changed to "monoterpene productions".

Page 3, line 8: remove the inner set of parentheses.

Response: Corrected.

Page 3, lines 13-15: This is a contentious statement, and there is no reference. A more nuanced statement is necessary, and could reference Monson, R.K., Grote, R., Niinemets, U., Schnitzler, J.P., 2012. Modeling the isoprene emission rate from leaves.New Phytologist 195, 541-559.

Response: We will add the description about process-based models can represent BVOC synthesis activities in chloroplasts and vary between species and leaf long-term growing environment. The suggested reference will be added.

Page 3, line 16: "referred to here" instead of "referred here"

Response: Accepted.

Page 3, line 17: remove space before comma

Response: Corrected.

Page 3, lines 18-20: Should reference Potosnak et al 2013 here. While dwarf willow's T response was OK compared to G93, the light response was more linear than expected.

Response: We will add this description. Thanks.

Page 3, line 25: Should also include low transpiration rates. Because of permafrost, transpiration rates can be low, which also leads to the high ground temperatures.

Response: Based on the observations, there is no permafrost at the studies plots. The issue is relevant for other arctic regions.

Page 3, line 30: Give what LPJ-GUESS stands for.

Reponses: the full name will be added.

Page 4, lines 2-4: The objectives could be clarified. To me, (1) "capture the observed BVOC T sensitivity" is the same as part of (2) "To address short-term and long-term impacts of warming on ecosystem BVOC emissions." Be more specific about your

study goals, or further differentiate the difference between 1 and 2.

Responses: Thanks for point out this part. The first aim will be clarified by changing it to "capture the observed T responses of BVOC emissions for a subarctic ecosystem", to clarify that we mainly tackle questions about emission responses to temperature. The second mainly aim to compare short-term and long-term warming effects on the whole ecosystem. We will clarify both aims.

Page 4, line 8: use straight single quote for minutes symbol.

Response: corrected.

Page 5, line 1: You have already defined PFTs above, so don't redefine.

Response: corrected.

Page 5, lines 2-4: Is this statement true for Arctic-specific PFTs? Please indicate this.

Response: Two of the cited references were conducted studies in high latitudes. So yes, this statement is true for arctic PFTs. As the reviewer suggested, the sentences will be clarified.

Page 5, line 5: What does "large-scale" mean here? I consider the base Farquhar equations to be leaf-level. Do you mean canopy-scale?

Response: The "large-scale" here mainly refers to the spatial scale. LPJ-GUESS uses the canopy-level photosynthesis calculation based on Haxeltine and Prentice (1996), where a set of canopy-level equations were developed from the Farquhar leave-level equations. We will clarify our description in the main text.

Page 5, lines 4-12: I assume transpiration & stomatal conductance are also modelled to get pi? Maybe you'll talk about this further down, but it would be important for understanding discrepancies between air and leaf temperature.

Response: Yes, in the model, the stomatal conductance influences transpiration as

well as intercellular CO2 concentration. We will clarify this in the descriptions.

Page 5, line 18: How is Ci different from pi defined on line 11. Just concentration vs. partial pressure? What does "without water stress" really mean? This is probably tied to my comment above.

Response: Thanks for pointing out. It should be pi and pi is influenced by stomatal opening, so we will correct both.

Page 5, line 26: "optimum from terpenoid synthesis" should be "optimum for terpenoid synthesis"

Response: corrected.

Page 6, lines 1-2: Give a reference for the co2 response in the model, as you've done for the other responses.

Response: corrected.

Page 6, lines 12-15: Again, this gets back to my comments above about transpiration and conductance. It would make more sense to move this discussion to the general description of the model, before discussing biogenics. Also, more detail on this part is necessary. What are the details here? This can be done by references to the literature, if it has been described by LPJ-GUESS before. What is the coupling between estimating leaf temp, internal co2, transpiration and stomatal conductance? Or is a more empirical algorithm used?

Response: Using leaf T instead of air T for photosynthesis was developed in this study, not in other LPJ-GUESS studies. The reason for putting the description of leaf T development after BVOC process description is that the development of leaf T algorithm mainly considers the strong sensitivity of BVOC to leaf T. We agree with the reviewer that the description of leaf T as well as its linkage to the transpiration and stomatal conductance should be extended. We will also stress the leaf T rather than air T was used for photosynthesis in this study.

Page 7, line 4: Fix grammar: either "appearing" or change sentence structure.ÂÍ

Response: changed.

Page 7, line 4-5: I agree there is insufficient data, but mosses may make a large contribution to BVOC emissions in some Arctic ecosystems. So, it's fine to incorporate them into a larger PFT, but are you capturing their emissions? That is, do the emission factors for this PFT reflex the mosses?

Response: The emission factors for the CLM have considered the observed moss emission rates. At the ecosystem level, we cannot distinguish how much emission was from the moss relative to the other species.

Page 7, line 17: first, not firstly.

Response: corrected.

Page 7, line 20: "other" instead of "rest"

Response: corrected.

Page 7, lines 18-21: Given the lack of data for the Arctic, it's justifiable to use two years data for calibration and two years for validation. But, the sensitivity of this procedure should be assessed by flopping the years: how different are the results if the second two years are used for calibration, and the first two used for validation?

Response: We did sensitivity testing using data from other two years (2010 and 2012) to calibrate, but this resulted only in slight effects on the best values for the checked LAI, GPP and ER, and the trend was consistent with that in Fig. S1. Further, it did not affect the selection of 0.04 $\mu$mol $CO_2$ $\mu$mol photons -1.

Page 8, lines 9-11: The goodness of fit here is a bit deceptive. The fit is entirely driven by the relatively few points that are above 23 deg C. Since everything below that is relatively close to zero, there is little new information added. For example, blocks 5 and 6 only have one observation each above 23 deg C, so the individual fits are very good. I

don't see the added value in the doing the individual fits for each block. It seems all that info comes out of the overall fit. Finally, you should understand the justification for using 20 instead of 30. Yes, this makes sense conceptually and certainly for measurements, but realizes that mathematically, using your formulation, there is no difference between using 20 and 30, because of the laws of exponents. That is, you'll get the same r2 for the fits with each. This isn't true with more complicated formulations of the T response; for example, the T response in isoprene emission for G93.

Response: The reason of adding block fitting is to illustrate the general fitting to the whole dataset also worked for each block (except for block 1) providing stronger evidence for the general trend. From the exponential equations along, we agree with the reviewer that there is no difference between using 20 or 30 degrees as reference temperature. However, the reference T in the model is not only used as BVOC T response equation, but also used in estimating photosynthesis electron flow rates at this reference T to convert the input emission capacity to fraction (see P7, line 35- P8, line1-2). The photosynthesis responses to the reference T of 20 and 30 degrees are not exponential. More explanation about the difference causing by different reference T will be added.

Page 8, lines 16-23: This is confusing. Your goal is to compare your measurements to the model. So, yes, using daily averages isn't appropriate. But why discuss them in the first place? I think you'll use them for another purpose, but that's not clear. Why do you use max T & PAR? Wouldn't an average around the measurement time make more sense? And again, your last sentence here is obvious. Particularly in the Arctic, with low sun angles for much of the day, this isn't a strong statement.

Response: We agree that the issue indeed was a bit confusing because of our wording. We will clarify that not all measurements were only between 10 am- 2 pm, but also with a few sampling between 9 am – 5 pm, and this is the reason why we still keep the daily average in the results. Since we don't use hourly inputs, it is not possible to average emissions for the time period of days with the measurements. We used theoretical

maximum T as the input for extracting daily maximum emission rates. Generally, it is not a very difficult problem to compute the maximum PAR, but we cannot really compute an instantaneous photosynthesis flux at noon (or any other time) with Haxeltine and Prentice approach. Lindwall et al. (2015) has shown strong diurnal cycle of BVOC emissions in the Arctic. This reference will be added to support our statement in the last sentence.

Page 8, lines 27-28: Again, examine Equation 3. You'll see that changing from 30 to 20 only introduces a constant.

Response: As explained in the previous responses, the reference temperature is not only used for the exponential equation (Eq. 3), but also used in LPJ-GUESS to link the photosynthesis rates at 20 degree.

Page 9, line 4: In Fig. S1, the figure legend should indicate what the dashed vertical line denotes at the value of 0.4 in both panels. The text explains this, but the figure caption should too.

Response: Corrected.

Page 9, line 10: Do you expect to see a one-to-one correspondence between the point intercept info and the LAI values? This surely doesn't hold as LAI gets closer to 1 (and exceeds it), but you should share your expectation here. Do you assume that there is no overlap with cover, and therefore there should be a one-to-one relationship? If so, state that.

Response: From the model side, LAI is the most relevant variable which can be used to compare with the point intercept measured plant coverage. The point intercept-based method does count numbers of plants that pin hits, not only the top canopy layer. So it should not have problem in comparing with LAI when LAI get larger or closer to 1. In this context, we did not assume no overlap with cover.

Page 9, lines 18-22: You discussed the LAI response to warming, but not the

GPP/NEP/ER response. Why?

Response: We did compare GPP response to warming as well and it showed that an underestimated vegetation CO2 fixation to warming in most cases, which can also been seen in the evaluating LAI response to warming (the absolute difference of total LAI between C and W plots). So we only included the LAI response to warming in the manuscript. Although the plant CO2 fluxes are also linked to BVOC emission, we consider the warming responses of LAI more relevant as they are aggregated effects of both photosynthesis responses and vegetation composition changes (directly linked to the changes in emitted compounds and relative magnitudes).

Page 9, line 29 – page 10, line 1: This analysis isn't adding much to your argument. Of course you see this, because your model is driven by PAR and T. You don't need to cover this result. It follows directly from your model formation (Equations 1-3). Second, I don't understand the relevance of relating mean daily ISO/MT production to the noontime values. What do you learn from this?

Response: Thanks for your points. We will move figure 4 in the supplementary instead. We think if reader is interested to see the seasonality of BVOC emissions in this region as a general picture about temporal dynamics of emissions, they can still reach it. As we explained in the earlier responses, due to the strong diurnal variations of BVOC emissions during a day, sampling time is crucial for determining the emission magnitudes. Although many samplings were conducted during the period of 10 am - 2 pm, there were also samplings conducted beyond this period. Through presenting both daily average and maximum values, we can conduct an approximate evaluation of how the model performs by checking if the measured rates were in the modelled range.

Page 10, lines 3-4: For "the observed average rates (blue squares) were well captured by the modelled noon emissions" you need to present some statistics to back up this statement. You should do an xy plot of this data and see what the fit looks like. Even if you don't present the plot as a figure, you should report the statistics of the fit.

Response: Thanks for pointing this out. We will add Willmott's index of agreement (A) as well as mean bias error (B) to describe the model's performance.

Page 10, line 10: As mentioned below, the same data is presented in Figs 4 and 5a. And now you've made the same statement about fit as above. This should be consolidated, and again there needs to be a statistical analysis of the goodness of fit.

Response: We will move Figure 4 and relevant descriptions into the Supplementary. So in this case, we will not have much data overlap in the figures. We will add statistics to the remaining figures.

Page 10, lines 13-18: Yes, the temperature drives these emissions, but this is a bit complicated because of the chamber observations. There are two issues: one, the model's ability to predict leaf T; second, the increase in air T because of the chamber used to measure BVOCs. Only the first is important for extrapolating your results.

Response: very good point. In this context, we can only discuss what we have considered in terms of leaf temperature estimation. However, the side effects of measurement chambers on leaf temperature were considered to be minor. As tested by De Boeck et al. (2012), the main side effects from chambers on leaf temperature is related to reduced wind speed. In our case, the measuring chamber has a fan to mix air during sampling time, in which we do not expect large impacts on the observed leaf temperature, but could have some impacts on chamber air temperature. However, for the warming treatment where the OTCs were installed to passively increase surrounding temperature, the OTC-resulted reduction of wind speed may elevate leaf T more than the expected on air temperature warming and could be considered as one reason for why the modelled WR was generally lower than the observed (Please see Section 4.2, first paragraph).

Page 10, lines 25-27: Again, need statistics to back up these contentions.

Response: We will add Willmott's index of agreement (A) as well as mean bias error
(B).

Page 11, line 30: After not using any statistics comparing the model to observations, why would you use a statistic in this case, when you are comparing the model to itself?

Response: When we evaluated the modelled emission rates at daily scale, we focused on presenting the absolute differences from the observed. But as the reviewer suggested, we will add statistics for the model-data comparison. For the modelled annual emissions, we don't have the observed data and to illustrate warming effects on the emissions, the Mann-Whitney test was applied.

Page 13, lines 23-25: This is an interesting contention. But, the emissions for the storage pool are generally regarded as being due to the physical process of evaporation of the MTs. Why would this change for Arctic plants?

Response: Parameterization of Eq. 4 is based on global scale study by Schurgers et al. (2009). It may be the case that for arctic plants, there is larger storage pools for MTs or different leaf anatomy which could influence release from storage pools. We are lacking of knowledge to quantify these effects on MTs emission, but we will clarify our discussion here.

Page 13, lines 28-29: Yes, and that is why restricting your modelling to the times of day when measurements occurred would help.

Response: Please see the previous response regarding to daily processes in the model.

Page 14, lines 2-3: Do you mean the bryophyte decrease due to drying is an artifact of the experimental warming and shouldn't be captured by the model? Please elaborate.

Response: The observations found an increase of bryophyte coverage, but our model predicted a decrease of the coverage. The decreasing trend in response to warming is consistent with the study by Elmendorf et al. (2012) where they summarized 61 tundra warming experiments. Elmendorf et al. (2012) elaborated that drying of soil moisture

is one of the reasons of declining of bryophyte coverage, which was captured by our model.

Page 14, lines 5-7: Yes, because you used the observed data to fit your model. Remind readers of that point.

Response: As mentioned in the earlier reply, we only used the emission rate and T at control plots to get the response curve, but compared the modelled WR with the observed, which has shown the improvement.

Page 14, lines 11-13: Could also mention the drying that was noted above for species responses (bryophytes).

Response: changed.

Page 14, lines 18-19: Need some more analysis here. Yes, the two responses are very important. But, you should re-emphasize that your dynamic vegetation model isn't doing a great job of getting the vegetation changes correct. Therefore, the results in Fig 8 are illustrative of the impact, but the details are not certain.

Response: Thanks for the good points. We agree with the reviewer that we only illustrate a potential impact, but that there are still uncertainties in capturing the vegetation dynamics in detail.

Page 14, lines 25-30: I agree with most of this logic, but since this particular study is looking at whole system measurements, there is potentially an interaction between the true T response of the plants and the issue of canopy temperature described earlier. You should at least discuss the possibility that some of this T response is not at the enzymatic level, as suggested here, but is due to a non-linear increase in leaf T with increasing air T due to canopy warming. Perhaps some of the references cited are leaf-level measurements which could clarify this point?

Response: The decoupling of leaf T from air T at 2 m height may partly contribute the observed strong warming responses. It may also relate to arctic plant species.

Discussion about the strong decoupling of leaf T from air T as well as its linkage to potential T response will be added.

Page 15, lines 1-3: Yes, this is important, but it also brings in the issue of drought stress. Drought stress can occur frequently in some Arctic ecosystems due to relatively shallow soils above the permafrost. To understand canopy heating, it will be necessary to understand canopy water dynamics.

Response: Thanks for the reviewer for pointing this out. In LPJ-GUESS, leaf energy balance has been considered and the evapotranspiration is a function of soil water content. For this particular site, there is no permafrost and the soil is relatively moist. As the reviewer correctly pointed out, drought stress for some other arctic regions are possible, which may lead to further increase in canopy surface temperature.

Page 15, lines 15-16: Great this is stated clearly in the conclusion, but this point should also be made in the discussion.

Response: changed.

Figures 2, 4, 5 and 6: For Figs 2, 5 and 6, use you DD/MM on the time axis, but day of year for Fig 4. Be consistent, and I prefer day of year.

Response: We will change to DD/MM for Fig. 4

Figures 5, 6: The top panels (a) of each figure are the same data presented in Figure 4. These results shouldn't be presented twice.

Response: We will only keep Figure 5 in the main text.

Figure 7: There is also a lot overlap with Figures 5b and 6b: two of the three sets of data have already been shown. In addition, why is there a break in the y-axis, when mostly the same data have been presented in Figures 5b and 6b without a break?

Response: I guess you mean Figure 6b and 7b. The reason for adding a break in y axis in Fig. 7 is the modelled WR from the simulation with the original T response is

really low, which is only presented here. In the revised version, we will use scatter plot to compare the differences between two T responses curves.

Figure 9: Why include the higher-T scenarios? I understand they are (unfortunately) realistic due to the IPCC estimates of climate change. But, you don't discuss them much, and there are obviously some weird things happening with the vegetation change (for example, lower +8 compared to +2 for MTs in 2012). Since the vegetation changes predicted by the model are suspect, the results of the +4 and +8 runs are highly speculative.

Response: The purpose of having Fig. 8 (Not figure 9) in the manuscript is to illustrate factors influencing BVOC emissions at long-term scale, highlighting that vegetation changes can affect the response at decadal timescales, and also illustrating that this can lead to a change in the response over time, or a non-linear response to the level of warming. We agree that these estimates are uncertain, also given the fact that the model does not capture all observed vegetation trends. We will expand the description of the results and discuss the underlying response.

De Boeck, H. J., De Groote, T., and Nijs, I.: Leaf temperatures in glasshouses and open-top chambers, New Phytologist, 194, 1155-1164, 2012.

Ekberg, A., Arneth, A., Hakola, H., Hayward, S., and Holst, T.: Isoprene emission from wetland sedges, Biogeosciences, 6, 601-613, 2009.

Elmendorf, S. C., Henry, G. H. R., Hollister, R. D., Björk, R. G., Bjorkman, A. D., Callaghan, T. V., Collier, L. S., Cooper, E. J., Cornelissen, J. H. C., Day, T. A., Fosaa, A. M., Gould, W. A., Grétarsdóttir, J., Harte, J., Hermanutz, L., Hik, D. S., Hofgaard, A., Jarrad, F., Jónsdóttir, I. S., Keuper, F., Klanderud, K., Klein, J. A., Koh, S., Kudo, G., Lang, S. I., Loewen, V., May, J. L., Mercado, J., Michelsen, A., Molau, U., Myers-Smith, I. H., Oberbauer, S. F., Pieper, S., Post, E., Rixen, C., Robinson, C. H., Schmidt, N. M., Shaver, G. R., Stenström, A., Tolvanen, A., Totland, Ø., Troxler, T., Wahren, C.-H., Webber, P. J., Welker, J. M., and Wookey, P. A.: Global assessment of experimental

climate warming on tundra vegetation: heterogeneity over space and time, Ecology Letters, 15, 164-175, 2012.

Haxeltine, A. and Prentice, I. C.: BIOME3: An equilibrium terrestrial biosphere model based on ecophysiological constraints, resource availability, and competition among plant functional types, Global Biogeochemical Cycles, 10, 693-709, 1996.

Lindwall, F., Faubert, P., and Rinnan, R.: Diel Variation of Biogenic Volatile Organic Compound Emissions- A field Study in the Sub, Low and High Arctic on the Effect of Temperature and Light, PLoS ONE, 10, e0123610, 2015.

Schurgers, G., Arneth, A., Holzinger, R., and Goldstein, A. H.: Process-based modelling of biogenic monoterpene emissions combining production and release from storage, Atmos. Chem. Phys., 9, 3409-3423, 2009.

---

## Author Comment (AC3) · 23 Jun 2016

This paper presents a very valuable and interesting work, focusing on isoprene and monoterpene emissions from subarctic plants, a topic that has not been investigated or published much so far. I really appreciate the originality of this study, which helps to improve our understanding regarding emission estimates. However, as also raised by the two other referees, I think that this manuscript would really benefit from a deeper and more detailed presentation, of the result analysis and discussion especially, which would help to appreciate more clearly the validity of the conclusions of this work. Here are some feedbacks and corrections that would need to be considered before publication in BG, that I warmly support.

Abstract: in "evaluating BVOC related processes", which processes for instance do you

refer to, photosynthesis?

Response: Here, we referred to photosynthesis, BVOC temperature responses and vegetation composition. The clarification in the abstract will be added.

Generally in the manuscript, the analysis is rather qualitative than quantitative and should be more detailed and specified. Some elements giving more precise information on the context could also be added. For instance, what is the estimated contribution of subarctic plants to global isoprene and monoterpene emissions? This could be specified for both the present-day case and the different warming scenarios, giving more perspective to the work carried out, and is important to be discussed, especially in section 4.

Response: Thanks for these good points. A statistical analysis of the model performance will be added. Regarding the contribution of subarctic plants' contributions to the global emissions, we would like to address in a coming manuscript where we will integrate multiple sites BVOC emission in the Arctic. We think it is a bit risk to estimate the contribution to global emissions based on one site study. We think the contribution to local atmospheric chemistry is potentially more important than the contribution to global emission (reactive compounds) and the warming-induced strong increase of emissions in this region is very important to address at global perspectives. We will add these two points in the Section 4.

Page 5, section 2.2.2 BVOC modelling: Could you please detail what the seasonality function used in isoprene production calculation stands for?

Response: The seasonality of isoprene production reflects observed changes in the availability of the enzyme for terpenoid synthesis, and is calculated based on a degree-day method in spring and a decrease in autumn based on temperature and day length. The details will be added.

Page 6, section 2.2.2 BVOC modelling: Works published so far agree on the CO2 inhibition effect regarding isoprene emissions, but not regarding other BVOC emissions. Is the f($CO_2$) function considered in the model only for isoprene or for every BVOCs? On which work is it based and is the same parameterization considered for every compounds?

Response: In the model, we used the same $CO_2$ response function for both isoprene and monoterpenes (and we do not use other BVOCs than those two) and the f($CO_2$) response is based on the work by Arneth et al. (2007). We assume in the model that isoprene and monoterpenes are produced in the same pathway and assume both responses to $CO_2$ in the same way. We agree with the reviewer that more work agreed on the $CO_2$ inhibition on isoprene. In the work by Peñuelas and Staudt (2010), they listed some studies (in the supplementary) with $CO_2$ inhibition effects on monoterpenes. We will extend our discussion on this topic and indicate this response is more robust for isoprene than monoterpenes.

Page 9, line 25: What do you mean exactly with "dynamic vegetation" in "simulating dynamic vegetation enables us to assess the model performance"? Day-to-day variability? Higher frequency? Indeed the term of dynamic vegetation can also refer in vegetation modeling to long-term changes in vegetation distribution due to climate and $CO_2$ changes.

Response: With the term "dynamic vegetation", we wanted to stress the model's ability to capture seasonal variations in leaf area as well as annual-decadal changes in vegetation composition. The sentence will be adjusted to clarify this in the revised manuscript.

The model/data comparison would also really benefit from a deeper analysis. If isoprene and monoterpene emission estimates fall into the data values, it is however difficult to come to a clear conclusion, as data are not that numerous, and as model estimates are given either as daily average or for noon. At what time were emission data collected and how are they compiled for model-data comparison?

Response: Thanks for the great points. Model-data comparison will be further analysed by adding statistics. For each data point, it is an average of six replicates in the field which were measured at different time points of a day (between 9 - 17). Since the model is running at daily time step, it is not possible to average the modelled emission rates at sampling time. We saw this limitation and therefore used the daily average and maximum as an indication about the model's performance.

The parameterization is calibrated and adjusted in order to better represent BVOC emission from Arctic plants. This is a crucial and one major contribution of this work and yet it is only very quickly mentioned in section 4.2. It is important to add a more detailed and quantitative analysis of the emission improvement, both in the results section and in the discussion part.

Response: Thanks for point out. We agreed with the reviewer that we should discuss the derived new temperature curve in a more detail. We will add more discussion regarding to parameterizing T response for arctic plants.

Specific comments:

Page 1, line 23: change "the model's responses" to "the model responses"

Response: changed.

Page 1, line 23: change "higher levels' warming" to "higher warming levels"

Response: changed to higher levels of warming

Page 2, line 4: change "Â′lPFT's reponses" to "PFT responses"

Response: changed to "physiological responses of PFTs"

Page 2, line 6: change "Biogenic volatile organic compounds (BVOC)" to "Biogenic volatile organic compounds (BVOCs)"

Response: Through the manuscript, we use BVOC as a plural term.

Page 2, line 11: change "atmosphere's oxidative capacity" to "atmosphere oxidative capacity"

Response: Changed.

Page 2, line 15: change "(. . . respectively (Sindelarova et al., 2014))" to "(. . . respectively; Sindelarova et al., 2014)"

Response: Changed.

Page 3; line 29: change "and to advance our understandings of the" to "and our understanding regarding"

Response: changed.

Page 3, line 30: remove comma in "ecosystem model, LPJ-GUESS"

Response: we will remove comma and add full name for the model.

Page 7, line 28: please change "LAI of the year 2006 and 2007" to "LAI of the years 2006 and 2007"

Response: changed.

Page 7, line 33, change "due to plants' adaptation" to "due to plant adaptation"

Response: changed.

Page 8, line 20-21: change "Due to lacking of data about the daily maximum" to "Due to the lack of data regarding the daily maximum"

Response: changed.

Page 9, line 26: change "to assess the model's performance" to "to assess the model performance"

Response: changed.

Page 15, line 11: change "the model's ability" to "the model ability"

Response: changed.

Figures:

It is hard to distinguish the observations from the emission estimates. Could you please trying using another color?

Response: We will modify the figures colors to make it contrast.

Arneth, A., Niinemets, Ü., Pressley, S., Bàck, J., Hari, P., Karl, T., Noe, S., Prentice, I. C., Serça, D., Hickler, T., Wolf, A., and Smith, B.: Process-based estimates of terrestrial ecosystem isoprene emissions: Incorporating the effects of a direct CO2-isoprene interaction, Atmospheric Chemistry and Physics, 7, 31-53, 2007.

Peñuelas, J. and Staudt, M.: BVOCs and global change, Trends in Plant Science, 15, 133-144, 2010.

---

## Author Response (AR1)

[revised manuscript text omitted]

Thanks for the reviewer's constructive comments and suggestions for improving this manuscript. All comments have been answered (with grey colour background). Below, we first give general answers to the reviewer's comments and the detailed comments will be addressed separately point by point.

Changes made in the revised manuscript have now been indicated (with green colour background). The page and line numbers mentioned in the changes refer to the ones in the revised manuscript.

Major comments

The subject matter of this paper is important. The Arctic environment is changing rapidly. Because of BVOC impacts on air chemistry, it's important to have models that can successfully predict the response of BVOC emissions. This paper makes an important contribution by employing a model with a dynamic vegetation component. As they warm, Arctic ecosystems are expected to see a shift towards woody plants, and this should change the capacity of the ecosystems to emit BVOCs. The paper has strengths, but also needs substantial improvements before publication. The basic modelling approach is sound, and it's helpful that the authors include the investigators that actually made the measurements. The paper demonstrates a good understanding of many of the ecosystem processes that should be captured by the model. Overall, I thought the discussion section was strong. Among weaknesses, the comparison of the model to the observations need to be improved. First, much of the discussion is qualitative. The model is said to fit the observations well in many instances, but there is no quantitative analyses: no goodness of fit metrics, and no statistics. Need to formally compare model to observations with statistics. More specifically, using the max and the daily average as a basis for comparison doesn't make much sense. What is the point of a daily average, especially since the meaning of the daily average changes with the long diurnal cycles in the Arctic? Why not just use the times of day that cover the range of the observations? Also, the figures could be improved by consolidation. The same data are presented in multiple figures in two different instances. The figures would also be easier to interpret if instead of presenting the max/daily average, just one metric was used for comparison to the observations. Also, there is very little acknowledgement of potential for experimental error in observations (one mention at the very end). Given the technical challenges with experiments in the Arctic, the potential for measurement error should be addressed.

The employed model is touted as being a mechanistic model, but then an empirical method is used for its calibration to the dataset. This is not itself a problem per se, but the paper states that mechanistic models are better than empirical models. If so, why is such an empirical calibration necessary? Also, a serious deficiency with the model is that it does not account for the effect of previous weather conditions (example, 24 hours and 10 days) on the capacity to emit BVOCs. This effect is potentially very important in the Arctic.

Finally, the list below of minor comments and technical corrections is extensive.

Response: (1) We agree with the reviewer that the comparison of modelled and observed variables should include some statistics and will add Willmott's index of agreement (describing models' prediction with pairwise-matched observations) as well as mean bias error (describing mean deviations between modelled and observed values) for each comparison.

Changes: The model's performances have now been evaluated by Willmott's index of agreement and mean bias error, see the descriptions in Page 10, Equation 5 and 6. These statistics have now been added in the results section.

(2) The reason why we present both daytime average and daily maximum values to compare with the observed is that the BVOC sampling on each field plot was conducted at a certain time point of a day for 30 min, while the modelled processes are at daily scale. Considering the strong diurnal cycle of BVOC emissions (for an arctic example, please see (Lindwall et al., 2015)), neither daily average nor daily maximum were accurate enough to directly compare with the observations. We reason that by presenting both rates, so we can see the range of the modelled daily emissions.

Changes: Both rates are still included in the manuscript, but clarification of the abovementioned reasons has now been added (see Page 9, lines 26-30).

(3) There is some data repetition in Figures 4-6, and separating them in different figures was aiming to explain different perspectives. In the revised manuscript, we will move Figure 4 to the supplementary to reduce data repetition. There will be no data repetition in Figure 5 and 6.

Changes: Figure 4 has been moved to the Supplementary Figure S2, together with previous description in the result section (see Page 11, lines 21-33 and Supplementary Page 6).

(4) We agree that the discussion about potential uncertainties from the measurements should be addressed in depth, mainly covering point intercept-based coverage and side effects from OTC-chambers.

Changes: The side effects of the OTC-chambers on warming responses were described in Page 17, Lines 5-7. The discussion about the point-intercept-based measurement was on Page 15, Lines 1-2. Discussion about the used technique for BVOC measurement has now been added on Page 16, Line 7-10.

(5) Regarding to the empirical method used in calibrating T response, we agree with the reviewer that there is empirical element in the parameter estimation, which, however, reflects some processes understanding, e.g. underlying enzyme activation. Even for mechanistic models in bio/geosciences fields, it sometimes cannot avoid of using empirical relationships determined from observations where multiple processes may anticipate. In our case, the calibrated T response (empirically) is used for influencing fraction of photosynthetic electron transport contributing to isoprene and monoterpene production, which is internally linked to other processes and potentially reflect more dynamics than other empirical models.

Changes: no changes have been made. Please see the above response.

(6) The current model did not consider the past weather conditions, rather, it emphasizes the enzymatic acclimation to short-term climate. As we do not have observed data to support which period of past weather should be taken into consideration, adding emission acclimation to the past weather will bring additional uncertainties to the modelled fluxes and complicate the current comparison. Ekberg et al. (2009) has fitted their observational data to obtain a relationship between past weather conditions (48 h) and isoprene emissions (not for monoterpene) from wetland sedges. We would like to further address this issue in the future model development with assistance of available climate data on measurement sites (past 24-96 h temperature as well as leaf level BVOC sampling data).

Changes: no changes have been made. Please see the above response.

Minor comments and technical corrections

Title: The title suggestions that the article will focus generally on modelling subarctic plants, but instead the article is about one specific effort using one specific model formulation. While of course some of the manuscript is more general, it is also uses data from just one field site.

Response: We can well understand the reviewer's concern about including only one study site and will therefore add a sub-title to specify it. The title will be changed to "Challenges in modelling isoprene and monoterpene emission dynamics of subarctic plants: A case study from a tundra heath".

Changes: New title: "Challenges in modelling isoprene and monoterpene emission dynamics of arctic plants: a case study from a subarctic tundra heath"

Page 1, line 14 – page 2, line 4: The abstract could be clearer. There are some specific recommendations below, but more generally the abstract should be condensed and just the highlights presented.

Response: Thanks for the comments. The abstract will be adjusted to condense the length and to make points clearer.

Changes: The abstract has been condensed by deleting a few sentences, See page 1, Lines 29-30 and Page 2, Lines 2-4.

Page 1, line 14: Title says "subarctic" while abstract goes back and forth between arctic and subarctic. Make sure each use is intentional. Further in the manuscript, (sub)arctic is used. Again, make sure this is all consistent.

Response: We will carefully consider each mention of "subarctic" and "arctic" through the whole manuscript. We will check and correct to use the most precise term in each place. The observational data originate at the Subarctic, but many ecosystem processes and components function similarly both in the Subarctic and the Arctic, and many of the issues handled are similar in both regions.

Changes: We have changed the title by stressing that the paper is presenting challenges for modelling BVOC in the Arctic based on this case study in a subarctic tundra.

Page 1, line 23: "higher levels of warming" instead of "higher levels' warming".

Response: Accepted.

Changes: Changed.

Page 1, lines 24-26: The sentence should be written. Do you mean the "measured" BVOC WR, not modeled? If you do mean modeled, what was the standard "were better captured"? Also, "compared" instead of "comparing".

Responses: Yes, it should be "measured". The suggested changes will be implemented.

Changes: Statistic has now been added in the main text (Page 12, lines 14-15).

Page 1, line 26: This sentence relays an interesting result, but there is not enough context to warrant inclusion in the abstract. Please remove it.

Response: The reason behind is the underestimated leaf T is one of the main factors influencing short-term BVOC fluxes. The sentence sounds a bit misleading, but we will clarify it.

Changes: The sentence has been changed to: A few days' underestimation of leaf T is one of the main causes responsible for the underestimated emission rates as well as WR.

Page 1, lines 30-31: This sentence can be removed, since it's a circular argument. The high WR led to the high adjustment T curve.

Response: The adjustment of T curve was only based on the emission rates from the control plots under naturally varying weather conditions within the growing season, and no emission rates from the warming plots were used. So, it is not high WR, which led to the high T curve. The improved ability of capturing the observed WR by the adjusted T curve indicates a better representation for arctic plants.

Changes: This sentence has been deleted to condense the abstract a bit.

Page 2, line 3: remove "extrapolation".

Response: Changed.

Changes: changed.

Page 2, lines 3-4: How do points (2) and (3) differ? Isn't "PTF's responses to warming" a subset of "representation of vegetation dynamics in the past and future"?

Response: we agree that these two points sound similar. Here, in point (2) "PFT's responses to warming", we mainly mean plant's physiological adaption to warmer climate, while in point (3) "representation of vegetation dynamics in the past and future", we mainly refer to long-term vegetation development, e.g., composition changes, disturbances, and expansion, etc. We will clarify the sentence.

Changes: Point 2 and 3 has been changed to: (2) PFT parameterization accounting for plant emission features as well as PFTs' physiological responses to warming; and (3) representation of long-term vegetation changes in the past and the future.

Page 2, line 7: "plant" instead of "plants" or include an apostrophe.

Response: Accepted.

Changes: changed

Page 2, lines 10-13: First, need to include that BVOCs don't solely react with OH. In particular, ozone is another important reaction partner for some BVOCs. Second, in a low-NOx environment, BVOC emissions can lead to a reduction in tropospheric ozone concentrations.

Response: The text will be clarified. Thanks for the points.

Changes:  The description has been changed to: An increase in BVOC emission could also elevate the tropospheric ozone ($O_3$) concentration when the ratio of BVOC to $NO_x$ (BVOC/$NO_x$) is high (Hauglustaine et al., 2005), and also increase secondary organic aerosol (SOA) formation (Paasonen et al., 2013). BVOC could also limit ozone formation when the BVOC/$NO_x$ ratio is low, a situation in which the regeneration of $NO_2$ can be mainly achieved by NO reacting with $O_3$ (Hauglustaine et al., 2005).

Page 3, line 3: "from" instead of "along". Also, why is G3P the "chief precursor" if pyruvate is also required?

Response: We have changed the words. The description of G3P as the chief precursor was not accurate and will be corrected.

Changes: The text has been changed to: both G3P and pyruvate serve as the chief precursor.

Page 3, line 6: "part of monoterpene productions" should be clarified.

Response: This is not accurate. It will be changed to "monoterpene productions".

Changes: Changed to "monoterpene productions".

Page 3, line 8: remove the inner set of parentheses.

Response: Corrected.

Changes: changed.

Page 3, lines 13-15: This is a contentious statement, and there is no reference. A more nuanced statement is necessary, and could reference Monson, R.K., Grote, R., Niinemets, U., Schnitzler, J.P., 2012. Modeling the isoprene emission rate from leaves.New Phytologist 195, 541-559.

Response: We will add the description about process-based models can represent BVOC synthesis activities in chloroplasts and vary between species and leaf long-term growing environment. The suggested reference will be added.

Changes: process-based ecosystem models, representing BVOC synthesis activities, can vary with species as well as long-term growing environment effects and could thus be more useful in terms of predicting long-term emission responses to environmental changes (Monson et al., 2012)

Page 3, line 16: "referred to here" instead of "referred here"

Response: Accepted.

Changes: changed.

Page 3, line 17: remove space before comma

Response: Corrected.

Changes: changed.

Page 3, lines 18-20: Should reference Potosnak et al 2013 here. While dwarf willow's T response was OK compared to G93, the light response was more linear than expected.

Response: We will add this description. Thanks.

Changes: The suggested descriptions is added: Potosnak et al. (2013) fitted leaf-level isoprene emission rates to T and Q in a moist acidic tundra and found the G93 algorithm characterized well with the T response, but not Q response.

Page 3, line 25: Should also include low transpiration rates. Because of permafrost, transpiration rates can be low, which also leads to the high ground temperatures.

Response: Based on the observations, there is no permafrost at the studies plots. The issue is relevant for other arctic regions.

Changes: Some discussions about other regions with permafrost underlying have been added on Page 17, Lines 2-4.

Page 3, line 30: Give what LPJ-GUESS stands for.

Reponses: the full name will be added.

Changes: The full name is: Lund-Potsdam-Jena General Ecosystem Simulator

Page 4, lines 2-4: The objectives could be clarified. To me, (1) "capture the observed BVOC T sensitivity" is the same as part of (2) "To address short-term and long-term impacts of warming on ecosystem BVOC emissions." Be more specific about your study goals, or further differentiate the difference between 1 and 2.

Responses: Thanks for point out this part. The first aim will be clarified by changing it to "capture the observed T responses of BVOC emissions for a subarctic ecosystem", to clarify that we mainly tackle questions about emission responses to temperature. The second mainly aim to compare short-term and long-term warming effects on the whole ecosystem. We will clarify both aims.

Changes: The text has been changed to: The specific objectives of this study were: (1) To capture the observed T response of BVOC emissions for a subarctic ecosystem; (2) To address the importance of short-term and long-term impacts of warming on ecosystem as well as BVOC emissions;

Page 4, line 8: use straight single quote for minutes symbol.

Response: corrected.

Changes: changed.

Page 5, line 1: You have already defined PFTs above, so don't redefine.

Response: corrected.

Changes: changed.

Page 5, lines 2-4: Is this statement true for Arctic-specific PFTs? Please indicate this.

Response: Two of the cited references were conducted studies in high latitudes. So yes, this statement is true for arctic PFTs.

Changes:  no changes have been made.

Page 5, line 5: What does "large-scale" mean here? I consider the base Farquhar equations to be leaf-level. Do you mean canopy-scale?

Response: The "large-scale" here mainly refers to the spatial scale. LPJ-GUESS uses the canopy-level photosynthesis calculation based on Haxeltine and Prentice (1996a), where a set of canopy-level equations were developed from the Farquhar leave-level equations. We will clarify our description in the main text.

Changes: The descriptions have now changed to "In LPJ-GUESS, a generalized Farquhar photosynthesis model (Farquhar et al., 1980; Collatz et al., 1991) for large-scale modelling is used to simulate canopy-level carbon assimilation and the generalized model is built on the assumption of optimal nitrogen (N) allocation in the vegetation canopy (Haxeltine and Prentice, 1996a; Haxeltine and Prentice, 1996b)"

Page 5, lines 4-12: I assume transpiration & stomatal conductance are also modelled to get pi? Maybe you'll talk about this further down, but it would be important for understanding discrepancies between air and leaf temperature.

Response: Yes, in the model, the stomatal conductance influences transpiration as well as intercellular $CO_2$ concentration. We will clarify this in the descriptions.

Changes: See page 5, lines 30-31 and page 7, lines 13-14.

Page 5, line 18: How is Ci different from pi defined on line 11. Just concentration vs. partial pressure? What does "without water stress" really mean? This is probably tied to my comment above.

Response: Thanks for pointing out. It should be $p_i$ and $p_i$ is influenced by stomatal opening, so we will correct both.

Changes: changed. See page 6, Equation 1 and lines 5-6.

Page 5, line 26: "optimum from terpenoid synthesis" should be "optimum for terpenoid synthesis"

Response: corrected.

Changes: changed

Page 6, lines 1-2: Give a reference for the co2 response in the model, as you've done for the other responses.

Response: corrected.

Changes:  The reference of Arneth et al., (2007) has been added.

Page 6, lines 12-15: Again, this gets back to my comments above about transpiration and conductance. It would make more sense to move this discussion to the general description of the model, before discussing biogenics. Also, more detail on this part is necessary. What are the details here? This can be done by references to the literature, if it has been described by LPJ-GUESS before. What is the coupling between estimating leaf temp, internal co2, transpiration and stomatal conductance? Or is a more empirical algorithm used?

Response: Using leaf T instead of air T for photosynthesis was developed in this study, not in other LPJ-GUESS studies. The reason for putting the description of leaf T development after BVOC process description is that the development of leaf T algorithm mainly considers the strong sensitivity of BVOC to leaf T. We agree with the reviewer that the description of leaf T as well as its linkage to the transpiration and stomatal conductance should be extended. We will also stress the leaf T rather than air T was used for photosynthesis in this study.

Changes: See changes from Page 7 lines 9-13.

Page 7, line 4: Fix grammar: either "appearing" or change sentence structure.¨

Response: changed.

Changes: changed.

Page 7, line 4-5: I agree there is insufficient data, but mosses may make a large contribution to BVOC emissions in some Arctic ecosystems. So, it's fine to incorporate them into a larger PFT, but are you capturing their emissions? That is, do the emission factors for this PFT reflex the mosses?

Response: The emission factors for the CLM have considered the observed moss emission rates. At the ecosystem level, we cannot distinguish how much emission was from the moss relative to the other species.

Changes: The measured emission rates from mosses had been integrated into the CLM group (see Table S2). No changes have been made.

Page 7, line 17: first, not firstly.

Response: corrected.

Changes: changed.

Page 7, line 20: "other" instead of "rest"

Response: corrected.

Changes: changed.

Page 7, lines 18-21: Given the lack of data for the Arctic, it's justifiable to use two years data for calibration and two years for validation. But, the sensitivity of this procedure should be assessed by flopping the years: how different are the results if the second two years are used for calibration, and the first two used for validation?

Response: We did sensitivity testing using data from other two years (2010 and 2012) to calibrate, but this resulted only in slight effects on the best values for the checked LAI, GPP and ER, and the trend was consistent with that in Fig. S1. Further, it did not affect the selection of 0.04 µmol $CO_2$ µmol photons $^{-1}$.

Changes: no changes have been made. Please see our explanation in the above response.

Page 8, lines 9-11: The goodness of fit here is a bit deceptive. The fit is entirely driven by the relatively few points that are above 23 deg C. Since everything below that is relatively close to zero, there is little new information added. For example, blocks 5 and 6 only have one observation each above 23 deg C, so the individual fits are very good. I don't see the added value in the doing the individual fits for each block. It seems all that info comes out of the overall fit. Finally, you should understand the justification for using 20 instead of 30. Yes, this makes sense conceptually and certainly for measurements, but realizes that mathematically, using your formulation, there is no difference between using 20 and 30, because of the laws of exponents. That is, you'll get the same r2 for the fits with each. This isn't true with more complicated formulations of the T response; for example, the T response in isoprene emission for G93.

Response: The reason of adding block fitting is to illustrate the general fitting to the whole dataset also worked for each block (except for block 1) providing stronger evidence for the general trend. From the exponential equations along, we agree with the reviewer that there is no difference between using 20 or 30 degrees as reference temperature. However, the reference T in the model is not only used as BVOC T response equation, but also used in estimating photosynthesis electron flow rates at this reference T to convert the input emission capacity to fraction (see P7, line 35- P8, line1-2). The photosynthesis responses to the reference T of 20 and 30 degrees are not exponential. More explanation about the difference causing by different reference T will be added.

Changes: Please see the added clarification at Page 6, line 14-15 and Page 9, line 5.

Page 8, lines 16-23: This is confusing. Your goal is to compare your measurements to the model. So, yes, using daily averages isn't appropriate. But why discuss them in the first place? I think you'll use them for another purpose, but that's not clear. Why do you use max T & PAR? Wouldn't an average around the measurement time make more sense? And again, your last sentence here is obvious. Particularly in the Arctic, with low sun angles for much of the day, this isn't a strong statement.

Response: We agree that the issue indeed was a bit confusing because of our wording. We will clarify that not all measurements were only between 10 am- 2 pm, but also with a few sampling between 9 am – 5 pm, and this is the reason why we still keep the daily average in the results. Since we don't use hourly inputs, it is not possible to average emissions for the time period of days with the measurements. We used theoretical maximum T as the input for extracting daily maximum emission rates. Generally, it is not a very difficult problem to compute the maximum PAR, but we cannot really compute an instantaneous photosynthesis flux at noon (or any other time) with Haxeltine and Prentice approach. Lindwall et al. (2015) has shown strong diurnal cycle of BVOC emissions in the Arctic. This reference will be added to support our statement in the last sentence.

Changes: see changes in Page 9, lines 26-31.

Page 8, lines 27-28: Again, examine Equation 3. You'll see that changing from 30 to 20 only introduces a constant.

Response: As explained in the previous responses, the reference temperature is not only used for the exponential equation (Eq. 3), but also used in LPJ-GUESS to link the photosynthesis rates at 20 degree.

Changes: Please see the added clarification at Page 6, line 14-15 and Page 9, line 5.

Page 9, line 4: In Fig. S1, the figure legend should indicate what the dashed vertical line denotes at the value of 0.4 in both panels. The text explains this, but the figure caption should too.

Response: Corrected.

Changes: changed.

Page 9, line 10: Do you expect to see a one-to-one correspondence between the point intercept info and the LAI values? This surely doesn't hold as LAI gets closer to 1 (and exceeds it), but you should share your expectation here. Do you assume that there is no overlap with cover, and therefore there should be a one-to-one relationship? If so, state that.

Response: From the model side, LAI is the most relevant variable which can be used to compare with the point intercept measured plant coverage. The point intercept-based method does count numbers of plants that pin hits, not only the top canopy layer. So it should not have problem in comparing with LAI when LAI get larger or closer to 1. In this context, we did not assume no overlap with cover.

Changes: No change has been made, but please see the discussion regarding to uncertainties in pin-point measurement (page 14, Line 33 to Page 15, line 1-2).

Page 9, lines 18-22: You discussed the LAI response to warming, but not the GPP/NEP/ER response. Why?

Response: We did compare GPP response to warming as well and it showed that an underestimated vegetation $CO_2$ fixation to warming in most cases, which can also been seen in the evaluating LAI response to warming (the absolute difference of total LAI between C and W plots). So we only included the LAI response to warming in the manuscript. Although the plant $CO_2$ fluxes are also linked to BVOC emission, we consider the warming responses of LAI more relevant as they are aggregated effects of both photosynthesis responses and vegetation composition changes (directly linked to the changes in emitted compounds and relative magnitudes).

Changes: no changes have been made.

Page 9, line 29 – page 10, line 1: This analysis isn't adding much to your argument. Of course you see this, because your model is driven by PAR and T. You don't need to cover this result. It follows directly from your model formation (Equations 1-3). Second, I don't understand the relevance of relating mean daily ISO/MT production to the noontime values. What do you learn from this?

Response: Thanks for your points. We will move figure 4 in the supplementary instead. We think if reader is interested to see the seasonality of BVOC emissions in this region as a general picture about temporal dynamics

of emissions, they can still reach it. As we explained in the earlier responses, due to the strong diurnal variations of BVOC emissions during a day, sampling time is crucial for determining the emission magnitudes. Although many samplings were conducted during the period of 10 am - 2 pm, there were also samplings conducted beyond this period. Through presenting both daily average and maximum values, we can conduct an approximate evaluation of how the model performs by checking if the measured rates were in the modelled range.

Changes: Figure 4 has been moved to Figure S2. The clarifications about why we present both daily noon and maximum values have been added in Page 9, lines 26-31.

Page 10, lines 3-4: For "the observed average rates (blue squares) were well captured by the modelled noon emissions" you need to present some statistics to back up this statement. You should do an xy plot of this data and see what the fit looks like. Even if you don't present the plot as a figure, you should report the statistics of the fit.

Response: Thanks for pointing this out. We will add Willmott's index of agreement (A) as well as mean bias error (B) to describe the model's performance.

Changes: See the added statistics on Page 12, lines 14-15.

Page 10, line 10: As mentioned below, the same data is presented in Figs 4 and 5a. And now you've made the same statement about fit as above. This should be consolidated, and again there needs to be a statistical analysis of the goodness of fit.

Response: We will move Figure 4 and relevant descriptions into the Supplementary. So in this case, we will not have much data overlap in the figures. We will add statistics to the remaining figures.

Change: Figure4 has been removed and statistics have been added to text and Figure 6.

Page 10, lines 13-18: Yes, the temperature drives these emissions, but this is a bit complicated because of the chamber observations. There are two issues: one, the model's ability to predict leaf T; second, the increase in air T because of the chamber used to measure BVOCs. Only the first is important for extrapolating your results.

Response: very good point. In this context, we can only discuss what we have considered in terms of leaf temperature estimation. However, the side effects of measurement chambers on leaf temperature were considered to be minor. As tested by De Boeck et al. (2012), the main side effects from chambers on leaf temperature is related to reduced wind speed. In our case, the measuring chamber has a fan to mix air during sampling time, in which we do not expect large impacts on the observed leaf temperature, but could have some impacts on chamber air temperature. However, for the warming treatment where the OTCs were installed to passively increase surrounding temperature, the OTC-resulted reduction of wind speed may elevate leaf T more than the expected on air temperature warming and could be considered as one reason for why the modelled WR was generally lower than the observed (Please see Section 4.2, first paragraph).

Changes: no changes have been made. Please see the explanation above.

Page 10, lines 25-27: Again, need statistics to back up these contentions.

Response: We will add Willmott's index of agreement (A) as well as mean bias error (B).

Changes: added. See changes on Page 13, lines 13-14.

Page 11, line 30: After not using any statistics comparing the model to observations, why would you use a statistic in this case, when you are comparing the model to itself?

Response: When we evaluated the modelled emission rates at daily scale, we focused on presenting the absolute differences from the observed. But as the reviewer suggested, we will add statistics for the model-data comparison. For the modelled annual emissions, we don't have the observed data and to illustrate warming effects on the emissions, the Mann-Whitney test was applied.

Changes: The added statistics have been indicated in previous answers.

Page 13, lines 23-25: This is an interesting contention. But, the emissions for the storage pool are generally regarded as being due to the physical process of evaporation of the MTs. Why would this change for Arctic plants?

Response: Parameterization of Eq. 4 is based on global scale study by Schurgers et al. (2009). It may be the case that for arctic plants, there is larger storage pools for MTs or different leaf anatomy which could influence release from storage pools. We are lacking of knowledge to quantify these effects on MTs emission, but we will clarify our discussion here.

Changes: see changes on Page 16, lines 2-6.

Page 13, lines 28-29: Yes, and that is why restricting your modelling to the times of day when measurements occurred would help.

Response: Please see the previous response regarding to daily processes in the model.

Changes: No changes have been made due to the modelled process at daily scale, but a change has been made about the measured time range (see Page 9, line 24).

Page 14, lines 2-3: Do you mean the bryophyte decrease due to drying is an artifact of the experimental warming and shouldn't be captured by the model? Please elaborate.

Response: The observations found an increase of bryophyte coverage, but our model predicted a decrease of the coverage. The decreasing trend in response to warming is consistent with the study by Elmendorf et al. (2012) where they summarized 61 tundra warming experiments. Elmendorf et al. (2012) elaborated that drying of soil moisture is one of the reasons of declining of bryophyte coverage, which was captured by our model.

Changes: no changes have been made. Please see the explanation above.

Page 14, lines 5-7: Yes, because you used the observed data to fit your model. Remind readers of that point.

Response: As mentioned in the earlier reply, we only used the emission rate and T at control plots to get the response curve, but compared the modelled WR with the observed, which has shown the improvement.

Changes: no changes have been made. Please see the explanation above.

Page 14, lines 11-13: Could also mention the drying that was noted above for species responses (bryophytes).

Response: changed.

Changes: added on Page 17, line 6.

Page 14, lines 18-19: Need some more analysis here. Yes, the two responses are very important. But, you should re-emphasize that your dynamic vegetation model isn't doing a great job of getting the vegetation changes correct. Therefore, the results in Fig 8 are illustrative of the impact, but the details are not certain.

Response: Thanks for the good points. We agree with the reviewer that we only illustrate a potential impact, but that there are still uncertainties in capturing the vegetation dynamics in detail.

Changes: The uncertainties about capturing vegetation dynamics have now been added in page 17, Lines 14-15. We also emphasized the uncertainties from the emission responses to environmental variables (Page 15, Lines 16-21)

Page 14, lines 25-30: I agree with most of this logic, but since this particular study is looking at whole system measurements, there is potentially an interaction between the true T response of the plants and the issue of canopy temperature described earlier. You should at least discuss the possibility that some of this T response is not at the enzymatic level, as suggested here, but is due to a non-linear increase in leaf T with increasing air T due to canopy warming. Perhaps some of the references cited are leaf-level measurements which could clarify this point?

Response: The decoupling of leaf T from air T at 2 m height may partly contribute the observed strong warming responses. It may also relate to arctic plant species. Discussion about the strong decoupling of leaf T from air T as well as its linkage to potential T response will be added.

Changes: Discussion about leaf T decoupling from air T has been added on Page 16, Lines, 4-5. Also, the discussion about the potential effects of different time resolution in the measured data and observed data has been added on Page 16, Lines 15-15 and Page 17, Lines 25-27.

Page 15, lines 1-3: Yes, this is important, but it also brings in the issue of drought stress. Drought stress can occur frequently in some Arctic ecosystems due to relatively shallow soils above the permafrost. To understand canopy heating, it will be necessary to understand canopy water dynamics.

Response: Thanks for the reviewer for pointing this out. In LPJ-GUESS, leaf energy balance has been considered and the evapotranspiration is a function of soil water content. For this particular site, there is no permafrost and the soil is relatively moist. As the reviewer correctly pointed out, drought stress for some other arctic regions are possible, which may lead to further increase in canopy surface temperature.

Changes: discussion about potential drying in the arctic permafrost region has been added on Page 17, lines 2-4.

Page 15, lines 15-16: Great this is stated clearly in the conclusion, but this point should also be made in the discussion.

Response: changed.

Changes: see discussions on Page 17, lines 14-16 and page 15, line 16-17.

Figures 2, 4, 5 and 6: For Figs 2, 5 and 6, use you DD/MM on the time axis, but day of year for Fig 4. Be consistent, and I prefer day of year.

Response: We will change to DD/MM for Fig. 4

Changes: see Figure S2.

Figures 5, 6: The top panels (a) of each figure are the same data presented in Figure 4. These results shouldn't be presented twice.

Response: We will only keep Figure 5 in the main text.

Changes: Figure 4 has been removed from the main text.

Figure 7: There is also a lot overlap with Figures 5b and 6b: two of the three sets of data have already been shown. In addition, why is there a break in the y-axis, when mostly the same data have been presented in Figures 5b and 6b without a break?

Response: I guess you mean Figure 6b and 7b. The reason for adding a break in y axis in Fig. 7 is the modelled WR from the simulation with the original T response is really low, which is only presented here. In the revised version, we will use scatter plot to compare the differences between two T responses curves.

Changes: A new scatter plot has been added to replace the original Figure 7.

Figure 9: Why include the higher-T scenarios? I understand they are (unfortunately) realistic due to the IPCC estimates of climate change. But, you don't discuss them much, and there are obviously some weird things happening with the vegetation change (for example, lower +8 compared to +2 for MTs in 2012). Since the vegetation changes predicted by the model are suspect, the results of the +4 and +8 runs are highly speculative.

Response: The purpose of having Fig. 8 (Not figure 9) in the manuscript is to illustrate factors influencing BVOC emissions at long-term scale, highlighting that vegetation changes can affect the response at decadal timescales, and also illustrating that this can lead to a change in the response over time, or a non-linear response to the level of warming. We agree that these estimates are uncertain, also given the fact that the model does not capture all observed vegetation trends. We will expand the description of the results and discuss the underlying response.

Changes: The purpose of having higher-T scenarios had been described in Section 2.3.4 (Page 10, Lines 12-13). Discussion regarding to the annual emission estimates has been added on Page 15, Lines 16-21. In the results section, we have addressed that vegetation is responsible for very different emission responses at 4 °C or 8 °C warming.

This is a nicely written manuscript which addresses an important question in BVOC estimation – namely the representation of cold environments in global estimates and the uncertainties of modelling in this respect. It is also well timed since a lot of new information has recently been published about this topic and the implementation of this knowledge into a model is overdue. However, I feel like I have to urge the authors to be more careful in what they regard as 'good agreement' between measurement and simulation or at which point they conclude that the model's suitability has been 'demonstrated'. Overall, I see a lot of model deficiencies and uncertainties in this study which should probably be the prime focus of the investigation. In this respect, I would welcome figures or statistics that show the actual relation between measurements and simulations rather than column- or point diagrams. Apart from this, I think that the model description part needs some elaboration.

Response: Thanks for the reviewer's suggestion. Briefly here, we will address the model's agreement with observations using a Willmott's index of agreement as well as mean bias error. Apart from BVOC related processes, a description of general photosynthesis processes will be added to Section 2.2.

Changes: Statistics have been added in the results section (see Page 12, lines, 14-15 and lines 17-18) and the model description has been extended in the section 2.2.1 (see Page 5, lines 16-21).

Specific comments:

P1, L22: 'Short time scales' not only need to be defined, mentioning them here is also irritating. In fact, the question about simulations and observations referring to different time periods is troubling me throughout the manuscript.

Response: The term refers to a period of hours to a few days versus long-term scale of months to years. The clarification will be added in the main text. Since the simulated results from the model were at daily scale and the measured fluxes could be at any time point of a day, presenting the modelled daily average and maximum values aimed to bridge the differences in the time periods.

Changes: The definition of different scales has been clarified in the abstract as well as in the main text. The reason why we present both daily maximum and average has been clarified on Page 9, Lines 28-30.

P1, L24: The model 'was able' to reproduce carbon fluxes for the majority of the vegetation period but showed considerable weakness in representing the seasonality, probably due to mismatch of phenological phases. This should be recognized.

Response: The modelled $CO_2$ fluxes do show some uncertainties in representing fluxes at the beginning of growing reason, which is discussed (see P13, L6-8) and related to phenological phases (the start of growing season). We agreed with the reviewer and will add the time period when the model did captured the observation. Also, we will change the term "was able to" to "showed reasonable agreement to".

Changes: The abovementioned changes have been added in the abstract. See Page 1, Line 26.

P1, L26: The difference of effective temperature in model and observation is certainly one reason for a mismatch in emission simulations which has been correctly acknowledged here. However, giving this as the only reason for a possible deviation is misleading at this point.

Response: Thanks for point out. The sentence will be clarified by stressing that leaf T is one potential main cause, but not the only reason for mismatches between model and observations.

Changes: The sentence has been deleted to condense the abstract.

P2, L17ff: Major uncertainties are also other driving factors for emissions that are usually not considered in models, namely air chemistry, soil water availability, UV light and biological stress impacts. Also the representation of seasonality (which is composed of phenology and enzymatic activity changes) is a point worth mentioning here. The authors are mentioning most of these points at a later stage but I feel that it needs mentioning here.

Response: Thanks for the great point. More details will be added in the introduction, paragraph 2.

Change: The suggested uncertainties have been added on Page 2, Lines 29-30.

P3, L5: I think that in the Pacifico and Unger papers, the Niinemets approach is used. So this is to some degree a repetition here.

Response: Agreed, we have reduced the references to unique implementations.

Changes: Two references: Pacifico et al., 2011 and Unger et al., 2013 have been removed.

P3, L10: seasonality and/or past weather conditions? In fact this is the same problem. You might differentiate into effects of phenology and enzymatic activity shifts though.

Response: We will change "seasonality" to "vegetation phenology" to differentiate relatively short-term acclimation (past weather condition) with vegetation phenological phases.

Changes: "Seasonality" has been changed to "vegetation phenology".

P4, L15: From the later remarks I take it that the BVOC emissions were not taken round the clock so the time or time period during the day when the measurements were made should be mentioned.

Response: A detailed description about measuring time will be added into the Section 2.3.3.

Changes: The update on the measuring time has been added on Page 9, line 24.

P5, L7ff: I am a bit irritated here. The Haxeltine and Prentice photosynthesis approach is for seasonal or annual photosynthesis estimation, assuming a kind of optimal adjustment to average environmental conditions. Nevertheless, the model seems to work on daily timesteps here. The description given about the model itself looks very much like the Collatz approach – so what is taken from Haxeltine here? Regarding the description, many abbreviations are introduced here that seem not to be used later on –please check.

Response: Thanks for pointing this out. We agree that the description (mainly references) of the photosynthesis processes was unclear. Though Haxeltine and Prentice model use monthly data as input, but it still have daily time step photosynthesis processes, which is what LPJ-GUESS is based on. The original simplified Farquhar model used in Haxeltine and Prentice is developed by Collatz et al. (1991) approach which works at sub-daily

scale. The model upscaling of leaf-level calculation to canopy scale is based on the Haxeltine's approach. The abbreviations which are not used later on will be deleted.

Changes: The descriptions have now changed to "In LPJ-GUESS, a generalized Farquhar photosynthesis model (Farquhar et al., 1980; Collatz et al., 1991)  for large-scale modelling is used to simulate canopy-level carbon assimilation and the generalized model is built on the assumption of optimal nitrogen (N) allocation in the vegetation canopy (Haxeltine and Prentice, 1996a; Haxeltine and Prentice, 1996b)"

P5, L14ff: Since emissions depend on temperature in a highly non-linear fashion, I think it is generally acknowledged that calculating them with daily average values is necessarily not capturing the dynamics. Regarding the Niinemets model, for example Unger et al. used a 15 minutes time steps. From the description it sounds like LPJ feeds daily photosynthesis results into daily emissions. Can you elaborate on the problem? Also, I think that the reference temperature used in equation 3 and/or the parameter in the response function needs to be adjusted because the model is not using them as an immediate response value anymore but as parameter for daily average emission. (30 degrees as an average value throughout the day would probably exhaust the emission apparatus so that the response curve would not be valid anyway.)

Response: Thanks for the good points. The simulations in this manuscript used daily climate inputs and therefore the model works on daily scale, resulting in daily emissions. To overcome (the largest part of) the problem rightly raised by the reviewer, we compute a daytime mean (rather than daily mean) temperature to simulate BVOC emissions (details in (Arneth et al., 2007)). This will be stressed in the revised manuscript. Still, the reviewer is correct that an average daytime temperature may still yield an underestimation of the emissions with the convex shape of the temperature response, certainly if the temperature variations during daytime are large. We will add discussion on this problem. To make our outputs comparable to a few time points measurements during a day, we came up this idea of presenting both daytime average and also daily maximum emission rate.

About the fitted curve with reference temperature of 20 degree, we are now aware of potential uncertainties caused by different time resolutions. In an ideal case, if we have more frequent BVOC samplings in a day as well as in the main growing season, we could average daytime T and emission rates before do the curve fitting. However, the current dataset is too few to support us to implement this parameter adjusting. The reviewer is correct and we will address this issue in the discussion at well. Thanks!

Changes: As explained above, the mismatches between of time resolution between the modelled and the measured cannot be completely solved due to the daily scale applied in the model. So no changes have been in this part. However, the daytime temperature, instead of daily average temperature was used in the model, and further clarified in the revised manuscript (see changes in Page 6, Line 19-20).

As above-mentioned reasons, we cannot adjust our temperature curve based on daily averaged data, due to the limited data availability.  But we are aware of potential uncertainties caused by different temporal resolution, and related discussion has therefore been added, see changes on Page 16, lines 15-18.

P5, L15: Instead of using I for isoprene as well as monoterpenes shouldn't you use Ei and Em or similar? This can further be modified for storage (e.g. Ems) in equation 4.

Response: The equations will be modified based on the suggestions.

Changes: The suggested symbol has been used, see Equation 1 and 4 on Pages 6-7.

P5, L22: Here, the influence is named 'phenology' while later the same function refers to 'seasonality' (L30). Since these are two different things – is this a lumped index? Specific or specifically parameterized for PFTs? Empirical or dependent on weather or climate?

Response: The use of "phenology" here is indeed not correct, $f(\sigma)$ represents the seasonality of the emissions caused by variations in enzyme activity. The effect of phenology (represented in the model as the abundance of leaves) is captured separately by affecting the amount of absorbed radiation. We will correct the sentence.

Changes: Please see the corrections on Page 6, line 5 and line 16. The model description regarding to plant phenology has been added on Page 6, line 9 and lines 22-23.

P5, L27ff: see also comment from L14ff. It seems that the reduction of reference temperature is rather a necessity from applying the model on a daily time step than a particular feature of arctic plants.

Response: Applying the reference temperature of 20 °C is of relevance for arctic plants since in most cases, the daytime T is close to or below 20 °C. We used the measured hourly BVOC fluxes with temperature in July to get the fitted temperature curve (see Fig. 1). The fitted response has been directly used in the model.

Changes: The clarification of using 20 °C for also computing photosynthesis fluxes has been added on Page 6, Lines 14-15, Page 8, Line 5. Potential uncertainties brought by different temporal resolution have been added on Page 16, line 15-18.

P5, L29: it is stated that the reference temperature is changed. This is to 20 oC as elaborated on later, correct?

Response: yes, we used the reference temperature of 20 °C. This will be clarified in the text.

Changes: See page 9, the 1st paragraph under Section 2.3.3.

P6, L2: fCO2 according to? Since it seems that variable CO2 air concentrations are used, it would be helpful to know to which degree CO2 might be responsible for differences between the years (probably small, but anyhow).

Response: Yes, the changes are small indeed, as f(CO2) varies with the inverse of the CO2 concentration. This gives a reduction of ~3% between 2006 and 2012.

Changes: see changes on Page 9, Line 11.

P6, L14: If the energy balance calculation was modified specifically for this study and is not published elsewhere, this modification should be explained.

Response: The development we had in this manuscript was essentially based on the work by Sedlar and Hock (2009) and therefore we did not include more details than just citing the original paper. But we will add more details about what are the main effects of adjusting the longwave radiation calculation.

Changes: The details have been added on Page 7, line 3-6. "The existing leaf energy balance equations appeared to underestimate the incoming longwave radiation under overcast conditions, which has been updated by

specifically considering the cloud emission of longwave radiation relative to clear-sky condition (Sedlar and Hock, 2009) The estimated leaf T, rather than air T, was used for both photosynthesis and BVOC synthesis."

P8, L15: I don't get how this can give you LAI values. Could you elaborate a bit? Looking at figure 3 there seems to be a difference between Lai and what is measured but the measurements are nevertheless used for evaluation. So how are the two related?

Response: The point intercept-based measurement gives a description of plant coverage (Finzel et al., 2012). During the growing season, the chances that the pins hit on leaves are generally higher and therefore we link these measured data with LAI which describes leaf coverage per ground area. It is not one-to-one relationship to compare (influenced by sampling inclining angles, sampling time, hits on stems etc., see discussion in section 4.1), but we think the modelled LAI is the closest variable we can compare with the measurement.

Changes: See the above explanation. No change has been made.

P8, L21: I agree that model results in daily resolution might not be comparable to measurements done at noon. This seems to be a general problem as mentioned above. I also agree that you can calculate noon temperature from average temperature to get a representative value of noon emission – but why don't you do the same with PAR? Instead of using the average value which is definitely wrong you can estimate maximum PAR from average PAR (e.g. Berninger F (1994) Simulated irradiance and temperature estimates as a possible source of bias in the simulation of photosynthesis. Agric. Forest Meteorol. 71:19-32)? Have you estimated the sensitivity of this error on the results?

Response: Thanks for the great point. Generally, it is not a very difficult problem to compute the maximum PAR, but we cannot really compute an instantaneous photosynthesis flux at noon (or any other time) with Haxeltine and Prentice approach, because it describes daily photosynthesis. It also becomes difficult to estimate potential sensitivity from different PAR values.

Changes: Please see the explanation above. Clarification has been made on Page 9, lines 26-27.

P9, L3: Check wording. I think it should be the modelled co2 fluxes that are sensitive to a change of parameter. This should also be indicated in some kind of measure, i.e. the degree to which the parameter was varied.

Response: The wording will be altered to clarify the text. The range for the parameter $\alpha_{c3}$ was based on a previous study by Pappas et al. (2013) and the changes of modelled $CO_2$ fluxes as well as LAI, responding to the parameter $\alpha_{c3}$ were illustrated in Figure S1. From Fig. S1, we can clearly see how the modelled CO2 and LAI varied with the parameter $\alpha_{c3}$. Before running sensitivity testing of $\alpha_{c3}$, we have selected several parameters to do sensitivity testing and then estimate Sobol sensitivity index to quantify the explained ability of each parameter to the modelled $CO_2$ fluxes as well as LAI.

Changes: Please see the changed wording on Page 10, Line 16-18. The pre-test/measure of varied degree of each parameter was tested but not presented.

P9, L9ff: In fact, the deviations are considerable. Not only GPP and thus emission is considerably overestimated in both years early seasons – which should be quantified and considered in annual estimates – but LAI is totally wrong in all PFTs except LSE+EPDS and CLM under current climate where the overestimation is a mere 10-15

percent. In L15/16 it is stated that these are the most important PFTs but in the next sentence the other PFTs are described to have a 'large coverage'. Are there any numbers that I have missed that give an objective picture about the abundances?

Response: Since the $CO_2$ fluxes are not continuously measured, quantification of the overestimated $CO_2$ fluxes of early season in annual estimates is unfortunately not possible. Considering that the modelled LAI and the point intercept-based may be not one-to-one relationship, the relative abundance of different PFTs coverage was evaluated. The measured coverage can be influenced by hits on non-leaf parts, pin size, subjective judgement of species and sampling inclining angles (see Discussion 4.1). We agree with the reviewer that the wording was at times confusing, e.g. the words "dominated" and "large coverage" and we will correct it.

Changes: No changes have been made in terms of estimating the overestimated GPP & emissions in annual emission due to limited observations. The pointed confusing sentence has been changed to "The two most dominant vegetation groups in the C plots, forbs/lichens and evergreen shrubs, were captured by the model. However, the coverage of graminoids (GRT) and non-Salix-type deciduous shrubs (NSLSS) was underestimated by our model. " Since we mainly look at compare the relative abundance of the modelled PFT LAI with the observed, no absolute numbers were compared between the modelled and the observed.

P10, L5: Monoterpene emissions seem to be met particularly because measurements occurred mostly on days with low emissions (according to figure 4). This is a problem because the high simulated emissions practically lack evaluation that should be addressed. I can certainly imagine other ways of representation or statistical analysis that can be used to elaborate on the point.

Response: Thanks for pointing this out. We will add discussion on the potential lacking evaluation of high monoterpene emission rates. We will also add the statistics for the comparisons and Figure 7 has changed to scatter plot to illustrate the modelled and the observed WR.

Changes: Please see the added discussion on Page 15, Line 29-32.

P10, L10: Similarly, I have large difficulties agreeing that figure 5 supports the statement that isoprene emissions were mostly captured by the model.

Response: This sentence actually pointed out that model is doing fairly good job on describing day-to-day variations of isoprene emission, though still have some discrepancies in capturing absolute magnitudes for some days. We will change our wording here and add statistic to support our description.

Changes: The sentence has been changed to: The observed daily variations in isoprene emissions were generally captured by the model (Fig. 4). The statistic has been added on Page 12, Lines 14-15.

P11, L26ff: The simulated annual emissions include the largely wrong response of LAI as well as the wrong response in early season emission, right? Can the error somehow be estimated? I have the feeling that these calculations might be too far off to be considered here.

Response: As mentioned in an earlier response, the closed-chamber based CO2 fluxes were not continuous measurements. The concluded overestimated $CO_2$ fluxes during the early seasons were based on very few measured data points. To further consider their influence on the annual estimate is difficult without continuous

data support. The simulated annual estimate is uncertain considering the mismatch in LAI and early season $CO_2$ fluxes, and we will clearly point out the uncertainty in the revised manuscript. However, presenting annual emissions in this manuscript is to look at longer timescales despite the discrepancies found in the evaluations.

Changes: As we mentioned, it may not be one-to-one relationship between the modelled LAI and the point intercept-based coverage. Instead of comparing the absolute values between these two, we mainly focused on the the modelled and the observed relative abundance as well as their response to warming. In this way, the model did fairly good job. So no changes have been made, but some wordings. About uncertainties of annual emissions, we have added more discussions on Page 15, lines 16-23.

P12, L14ff: The discussion seems to be overall comprehensive. Still, as for example in the first line, I think the authors are overenthusiastic about their results. This also applies for the conclusions.

Response: we will adjust the wording.

Changes: The sentence has been changed to: The modelled day-to-day variations of ecosystem CO2 fluxes (Fig. 2) and BVOC emissions generally followed the observations. See page 14, lines 12-13.

P14, L23ff: The comparison with common parameterization should not only be concentrated on the arctic environment but also on the problem with the time resolution (see above).

Response: The time resolution could be a possible cause. As mentioned in an earlier reply, the model has used daytime temperature, instead of daily temperature, which could reduce potential differences caused by two time scales. We will add discussion about potential influences of time resolution on emission T response in Section 4.2.

Changes: The clarification of daytime temperature used in the model has been added on Page 6, lines 19-20. The discussion about uncertainties from different temporal resolutions has been added on Page 16, Lines 15-18.

Page 9, line 25: What do you mean exactly with "dynamic vegetation" in "simulating dynamic vegetation enables us to assess the model performance"? Day-to-day variability? Higher frequency? Indeed the term of dynamic vegetation can also refer in vegetation modeling to long-term changes in vegetation distribution due to climate and CO2 changes.

Response: With the term "dynamic vegetation", we wanted to stress the model's ability to capture seasonal variations in leaf area as well as annual-decadal changes in vegetation composition. The sentence will be adjusted to clarify this in the revised manuscript.

Changes: The sentence has been changed to: BVOC emissions are closely linked to leaf as well as ecosystem developments. Simulating vegetation seasonal variations in leaf area as well as vegetation compositions enables us to assess the model performance in representing short-term emission changes in response to T and PAR, as well as long-term changes in vegetation development and distribution.

The model/data comparison would also really benefit from a deeper analysis. If isoprene and monoterpene emission estimates fall into the data values, it is however difficult to come to a clear conclusion, as data are not that numerous, and as model estimates are given either as daily average or for noon. At what time were emission data collected and how are they compiled for model-data comparison?

Response: Thanks for the great points. Model-data comparison will be further analysed by adding statistics. For each data point, it is an average of six replicates in the field which were measured at different time points of a day (between 9 - 17). Since the model is running at daily time step, it is not possible to average the modelled emission rates at sampling time. We saw this limitation and therefore used the daily average and maximum as an indication about the model's performance.

Changes: Please see the added statistic equations 5 and 6 on Page 10 and the numbers in the result section (Page 12, lines 14-15 and Page 13, lines13-14). The clarification about why we use both daily average and noon have been added on Page 9, Lines 28-30.

The parameterization is calibrated and adjusted in order to better represent BVOC emission from Arctic plants. This is a crucial and one major contribution of this work and yet it is only very quickly mentioned in section 4.2. It is important to add a more detailed and quantitative analysis of the emission improvement, both in the results section and in the discussion part.

Response: Thanks for point out. We agreed with the reviewer that we should discuss the derived new temperature curve in a more detail. We will add more discussion regarding to parameterizing T response for arctic plants.

Changes: In the result section, the comparison between the original T curve and the new one has been quantified by using scatter plot (see Figure 6). In the discussion section, we have added some explanation and discussion of this strong T response (using the adjusted T curve) (Page 17, Lines 2-7). The associated high T during sunny days was explained in the one sentence before. Also, the uncertainties associated the data as well as method we used for deriving the T curve have been added on Page 17, Lines 25-27.

Specific comments:

Page 1, line 23: change "the model's responses" to "the model responses"

Response: changed.

Page 1, line 23: change "higher levels' warming" to "higher warming levels"

Response: changed to higher levels of warming.

Page 2, line 4: change "´lPFT's reponses" to "PFT responses"

Response: changed to "physiological responses of PFTs"

Page 2, line 6: change "Biogenic volatile organic compounds (BVOC)" to "Biogenic volatile organic compounds (BVOCs)"

Response:  Through the manuscript, we use BVOC as a plural term.

Page 2, line 11: change "atmosphere's oxidative capacity" to "atmosphere oxidative capacity"

Response: Changed.

Page 2, line 15: change "(. . . respectively (Sindelarova et al., 2014))" to "(. . . respectively; Sindelarova et al., 2014)"

Response: Changed.

Page 3; line 29: change "and to advance our understandings of the" to "and our understanding regarding"

Response: changed.

Page 3, line 30: remove comma in "ecosystem model, LPJ-GUESS"

Response: we will remove comma and add full name for the model.

Changes: the full name has been added: Lund-Potsdam-Jena General Ecosystem Simulator

Page 7, line 28: please change "LAI of the year 2006 and 2007" to "LAI of the years 2006 and 2007"

Response: changed.

Page 7, line 33, change "due to plants' adaptation" to "due to plant adaptation"

Response: changed.

Page 8, line 20-21: change "Due to lacking of data about the daily maximum" to "Due to the lack of data regarding the daily maximum"

Response: changed.

Page 9, line 26: change "to assess the model's performance" to "to assess the model performance"

Response: changed.

Page 15, line 11: change "the model's ability" to "the model ability"

Response: changed.

Figures:

It is hard to distinguish the observations from the emission estimates. Could you please trying using another color?

Response: We will modify the figures colors to make it contrast.

Changes: Figure 4 and 5 have changed colours and Figure 6 has changed from bar plot to scatter plot.

---

## Author Response (AR2)

**Dear Editor and Reviewers,**

We appreciate your time and efforts for commenting on this manuscript. As both the reviewers have correctly pointed out, the comparison between model and observations was not done in the best possible way. We have solved this issue in the revision with a recalculation of emissions with the actual conditions at the time of sampling.

The other points brought up by the reviewers were also addressed, and responses to the individual comments can be found below (with grey background). The page and line numbers mentioned in the replies refer to the ones in the revised manuscript (without marked-up changes). We hope to have addressed your comments satisfactorily.

Thanks again for your great contributions.

**Best regards, on behalf of all co-authors,**
**Jing Tang**

**Replies to the 1st reviewer:**

Although many comments and suggestions have been acknowledged and considered, I am sorry to say that I am not quite convinced about the argumentation not to change a thing about noon-calculations and LAI representation. This now refers to the following parts of the manuscript:

P6, L11: The seasonality function only applies to isoprene production although (L14) the model assumes that both isoprene and monoterpenes are produced in the same pathway and respond in the same way to CO2. This is an inconsistent approach. Either the seasonality should apply on monoterpenes too or the CO2 response cannot be applied on (light dependent) monoterpenes.

10  Reply: Thanks for pointing out this inconsistency. We have now revised our model and applied the same seasonality function to monoterpene production. The Materials and methods section has been edited correspondingly (P6, L9-10). The impact of this change on daily and annual monoterpene emissions appeared to be very small. It plays a role only at the beginning and the end of the growing season, when emissions are generally low. The seasonality function regulates emissions of deciduous

15  plants only, as is the case for isoprene.

It is not clear from the description but do coniferous and herbaceous PFTs light-dependent and light-independent fractions at the same time? I guess not because then I would like to know how the epsilon_s parameter is derived from measurements (Had there been an a-priori assumption about the differentiation? Have I missed this piece of information?). Please clarify in the text.

20  Emissions from storages should depend on average daily temperature, not on daytime temperature. Therefore, the T values in Eq. 3 and 4 should not be the same. Has this been considered?

Reply: In the model, BVOC synthesis uses a fraction of the photosynthetic electron flux (J in Eq. 1), and synthesized terpenoids are either entirely emitted directly (e.g. isoprene), or part of the production is stored. The latter is the case for coniferous and herbaceous PFTs, where 50% of the produced

25  monoterpenes are put into the storage pool, from which emissions are computed based on the pool size and a temperature dependence (Eq. 4). Simulated monoterpene emissions originate hence from a combination of light-dependent and light-independent sources.

The use of 50% to distribute between emission and storage originates from Schurgers et al. (2009), and similar values were obtained from $^{13}CO_2$ labelling by Ghirardo et al. (2010) for coniferous trees. We have not made an attempt to distinguish between light-dependent and light-independent emissions from the samples taken in this study; the strong correlation between temperature and light would require a different setup (e.g. measurements in darkness) to do this correctly. This remains a weakness of the current study, which has been addressed in the discussion (P14, L16-20).

The reviewer is correct that daily mean temperatures should be used for emissions from storage. In the model, this has actually been done correctly. We have now corrected Eq. 4 and the relevant text to show that daily temperature is used (P6, L6 and L16-18, Eq. 4).

P9, L4: I am not convinced from the response regarding using only average PAR for calculating emissions because 'the Haxeltine and Prentice approach only describes daily photosynthesis'. Why should it not be possible 'to compute an instantaneous flux at noon' all the more why can an artificially calculated flux (that might be empirically increased to a reasonable degree if not calculated) not be fed into the photosynthesis model to check its response? If this would be a reason, the use of maximum temperature to calculate an upper limit for emission would also not be valid.

Reply: We agree with the reviewer that our previous attempts to compare model and observations contained unrealistic assumptions, and that the use of daily maximum air T was not good enough. We have revised our model-data comparison now, and have come up with a better solution using measured T inside the enclosure and ambient PAR values at the time of sampling, rather than using the daily climate data. In the model, the measured canopy air T and PAR were only used for re-estimating the photosynthesis fluxes and BVOC emissions at the time of sampling (for direct comparison with the observations), and these computations did not have any impact on the long-term simulation of vegetation dynamics and daily/annual BVOC emission rates. The new inputs and methods have been described on P7, L7-8 and P9, L8-12. The old Figures 4 and 5 have now been updated and integrated into one (Fig. 4, P27) with the re-estimated emission rates and WR. The result section has been changed accordingly with the new estimation of daily emission rates and WR using the measured T and PAR (P11, L1-23).

In this process of revision of the model-data comparison, we made the decision to exclude the observed monoterpene emissions in 2010 from the current paper. This was done because of gaps in the monoterpene, enclosure temperature and ambient PAR data for these measurements. In 2010, technical problems had prevented analysis of isoprene (Valolahti et al., 2015), and we have decided to take a conservative approach and not use the corresponding monoterpene data either. This enables integration of Figs 4 and 5 into one, and similar sets of observational data for both isoprene and monoterpenes.

P10, L13/14: I don't see any valid response, explanation or consideration of the fact that the modelled and measured LAI values are far off for most of the vegetation types. In figure 3, the forbs/lichens type (CLM) LAI is about 0.5 measured and 0.25 modelled, the evergreen shrubs (LSE+EDPS) are presented with a measured LAI of almost 0.4 while the model gives 0.2 (note the different axis). So how can this be called 'captured by the model'?? Or was there a kind of mix up with the axis?

Reply: We can see that the large difference in measured vs. observed LAI was still poorly explained in the manuscript, and we have taken several measures to clarify and better acknowledge the issue. We can identify several reasons for the large discrepancy, which are explained below. The choice of different axis scales was not a mixed up, but a deliberate choice. They were chosen to allow for comparison of the effects of warming on the coverage of the different PFTs in the observed and modelled data. We can see that the reason was not clearly articulated, so this has been done now both where the figure is explained and in the figure legend (also see below).

In the Materials and methods, we explain the basic difference in LAI and the point-intercept-based coverage, i.e. that they "are not comparable one-to-one throughout growing seasons, since the measurement includes pin hits on different plant parts, whereas LAI only explains leaf coverage. However, the point-intercept-based coverage approaches leaf coverage when the deciduous leaves become fully developed during the growing season."

We have now edited the text in the Results (P10, L13-17) to clearly acknowledge the large underestimation of the PFT coverage by the modelled LAI compared to the observed point-intercept-based coverage. We have also added a reason for using the different axis scales "note different left and right axis scales in Fig. 3 to allow comparison of relative changes in response to warming" (P10, L14-15). This has also been amended in the legend of Fig. 3 to explain the use of different scales.

We also clearly acknowledge the mismatch in the discussion, already in the first sentence: "…in spite of the poor representation of the observed vegetation composition". We have edited the second sentence (P12, L26-29) to explain the most probable reason to the mismatch: "…LAI only includes the areal coverage by leaves, whereas the point intercepted-based vegetation coverage also includes coverage detected of other aboveground plant parts, like stems." The other reasons contributing to the difference (underestimation of the allocation of assimilated carbon to foliage in LPJ-GUESS and/or too low SLA values plus methodological issues in the point-intercept technique) are explained on P12, L29-31, P13, L1-5 and P13, L13-15, respectively.

In addition, you might consider to modify the following:
P7, L32: Since evaluation and validation are different terms, and what has been done here is clearly 'evaluation', this should not be mixed up in the headline (in short: change validation into evaluation). (Check also throughout the manuscript.)

Reply: Thanks for pointing this out. We agree and have now replaced all terms "validation" with "evaluation".

**Replies to the 2nd reviewer:**

I appreciate the many changes the authors have made to improve the manuscript in response to the two reviewers' comments, but I still have two major concerns. These were both concerns that I raised during my original review which I feel have not been adequately addressed.

First, and more importantly, I am still unsatisfied with the comparison of modelled daily means and maxes with the spot measurements. I see that my concerns were also shared by the other reviewer. I do appreciate that the authors have moved past the very qualitative comparison that was employed in the first version of the manuscript. But, the authors are using a model that produces only one estimate of BVOC emissions per day, and it's difficult to compare this to a relatively small number of measurements. My view is given the difficulty of acquiring the field data, these measurements are useful and I understand I can't demand the perfect dataset with more complete coverage. On the other hand, modeling is much more flexible, and more sophisticated modeling approaches can be applied. The authors can run a series of different simulations using different weather inputs. For example, they could apply a Monte Carlo approach that can give information about both mean responses and variance. This is a much more appropriate response than simply doing two runs with the mean and max values. I would be more satisfied in the authors could site a previous study that used only daily means and maxes in a similar comparison.

Reply: Thanks for the suggestions. We have now revised the comparison between model and data and have come up with a solution using observed air T in the enclosure and ambient PAR at the time of sampling to re-compute the emission at the time of sampling (see new descriptions in Methods section, P7, L7-8 and P9, L8-12 and new figure 4 on P27)., which is the most accurate representation of expected emissions. In the model, the measured air T inside the enclosure and PAR were only used for re-estimating the photosynthesis fluxes and BVOC emissions at the time of sampling, and these computations did not have any impact on the long-term simulation of vegetation dynamics and daily/annual BVOC emission rates.

Second, and I understand this is a more minor and somewhat picky point, but the authors should more carefully read Monson et al 2012, which they cite on page 3 of the new manuscript. In particular, see section XI (Conclusions) in that reference. The main point is that so-called 'mechanistic' models, based

on the Niinemets approach, have an empirical gap due to the lack of knowledge about a number of critical processes. Again, I encourage the authors to add more nuisance to their comparison of the modelling approaches. Since the authors are not bringing anything new to the table in terms of the comparison of mechanistic vs empirical models, I don't understand the necessity of disparaging empirical models. The authors can simply state what they are using, and move forward.

Reply: Thanks for pointing this out. We have now clarified what the model can simulate in a more mechanistic way (vegetation dynamics and long-term response) and what this means for simulating BVOC emission). We also took away the unnecessary comparison between different modeling approaches, see P3, L12-14.

**References**

Ghirardo, A., Koch, K., Taipale, R., Zimmer, I. N. A., Schnitzler, J.-P., and Rinne, J.: Determination of de novo and pool emissions of terpenes from four common boreal/alpine trees by 13CO2 labelling and PTR-MS analysis, Plant, cell & environment, 33, 781-792, 10.1111/j.1365-3040.2009.02104.x, 2010.

[revised manuscript text omitted]

**S3 Seasonal variation of BVOC emissions**

The span of the BVOC measurements covered the main growing seasons over three years. The modelled daily average emission rates in the C plots showed pronounced day-to-day and seasonal variations (Fig. S2). The modelled emissions of isoprene and monoterpenes were low in  Spring and Autumn, and peaked on warm days during the Summer. The day-to-day variations in the emissions agreed well with the variations of T and PAR. When both T and PAR were high , the peaks of both isoprene and monoterpene emissions occurred . The observed magnitude of isoprene emissions during daytime showed large spatial variation between the blocks for the days with the observed high average emission rates (blue error bars in Fig. S2 with low emissions The emission of monoterpenes remained more constant than that of isoprene towards the end of the growing season (not fully presented here).

[Figure]

[Figure]

**Figure S2 Time-series of the air temperature (Air T) at 2 m height, photosynthetically active radiation (PAR), the modelled isoprene (ISO) and monoterpene emissions (MT) for the days 150-250 in 2006, 2007, 2010 and 2012 in the Abisko tundra heath. Both modelled and observed fluxes are from the control (C) conditions and the modelled daily average (Mod-C: daily average) and daily noon (Mod-C: noon) emissions are presented. Error bars indicate the standard deviation for the six replicates. For the year 2010, isoprene emission rates were not analyzed due to technical problems.**